# A flexible loop in the paxillin LIM3 domain mediates its direct binding to integrin β subunits

**Timo Baade**[1,2☯], **Marcus Michaelis**[3,4☯], **Andreas Prestel**[3¤], **Christoph Paone**[1,2], **Nikolai Klishin**[3,4], **Marleen Herbinger**[1], **Laura Scheinost**[1], **Ruslan Nedielkov**[3], **Christof R. Hauck**[1,2]*, **Heiko M. Möller**[3,4]*

**1** Lehrstuhl Zellbiologie, Universität Konstanz, Konstanz, Germany, **2** Konstanz Research School Chemical Biology, Universität Konstanz, Konstanz, Germany, **3** Analytische Chemie, Universität Potsdam, Potsdam, Germany, **4** DFG Research Training Group 2473 "Bioactive Peptides"

☯ These authors contributed equally to this work.
¤ Current address: Section for Biomolecular Sciences, The Kaj Ulrik Linderstrøm-Lang Centre for Protein Science, Structural Biology and NMR laboratory, Copenhagen, Denmark
* christof.hauck@uni-konstanz.de (CRH); heiko.moeller@uni-potsdam.de (HMM)

**Data Availability Statement:** The resonance assignment of paxillin LIM2/3 has been deposited to the BMRB (Entry# 51154). The coordinates of the final ensemble of Paxillin LIM2/3 domains have

## Abstract

Integrins are fundamental for cell adhesion and the formation of focal adhesions (FA). Accordingly, these receptors guide embryonic development, tissue maintenance, and haemostasis but are also involved in cancer invasion and metastasis. A detailed understanding of the molecular interactions that drive integrin activation, FA assembly, and downstream signalling cascades is critical. Here, we reveal a direct association of paxillin, a marker protein of FA sites, with the cytoplasmic tails of the integrin β1 and β3 subunits. The binding interface resides in paxillin's LIM3 domain, where based on the NMR structure and functional analyses, a flexible, 7-amino acid loop engages the unstructured part of the integrin cytoplasmic tail. Genetic manipulation of the involved residues in either paxillin or integrin β3 compromises cell adhesion and motility of murine fibroblasts. This direct interaction between paxillin and the integrin cytoplasmic domain identifies an alternative, kindlin-independent mode of integrin outside-in signalling particularly important for integrin β3 function.

## Introduction

Integrins are specialized cell surface receptors of animal cells that sense the extracellular matrix and coordinate cell adhesion with the organization of the cytoskeleton [1]. Integrins are transmembrane glycoproteins consisting of an α and β subunit, with 18 distinct integrin α and 8 integrin β subunits encoded in the human genome [2]. The high affinity, active conformation of these heterodimers can be stabilized either by extracellular ligand binding (outside-in signalling) or by distinct intracellular signalling processes, which allow the association of the cytosolic scaffolding proteins talin and kindlin with the β subunit cytoplasmic tail (inside-out signalling) [3]. Active integrins, together with their binding partners talin and kindlin, serve as the nucleus for the initiation of large, multimeric protein complexes termed "focal adhesions"

been deposited to the PDB (Accession code: 7QB0). The ImageJ macro used to quantify the cell spreading data has been deposited to the Zenodo database (https://zenodo.org/doi/10.5281/zenodo.12736436).

**Funding:** Funding for this work was awarded to CRH via CRC969, project B06 by Deutsche Forschungsgemeinschaft (https://www.dfg.de/). The work and position of TB, CP and MH was supported by the CRC969. Funding and support for this work was awarded to HMM, MM, and NK via RTG 2473 (Project number 392923329), project C3 by Deutsche Forschungsgemeinschaft (https://www.dfg.de/). The work of MM was supported by the RTG 2474. The work and position of NK was funded by the RTG 2474. The funders had no role in study design, data collection and analysis, decision to publish, or preparation of the manuscript.

**Competing interests:** The authors have declared that no competing interests exist.

**Abbreviations:** CEACAM, Carcinoembryonic antigen-related cell adhesion molecule; CSP, chememical shift perturbation; CT, carboxy-terminal; FA, focal adhesion; FAK, focal adhesion kinase; FCS, fetal calf serum; GFP, green fluorescent protein; HRP, horseradish peroxidase; IPTG, isopropyl β-Dthiogalactoside; ITGB, integrin beta; KO, knockout; LIC, ligation independent cloning; LIM, Lin-11, Isl1, MEC-3; MEF, mouse embryonic fibroblast; NMR, nuclear magnetic resonance; NOE, nucelar Overhauser effect; OPTIC, Opa protein triggered integrin clustering; PCR, polymerase chain reaction; PXN, paxillin; RFP, red fluorescent protein; SUMO, small ubiquitin-like modifier; WCL, whole cell lysate; wt, wild type.

(FAs) [4,5]. Combined biochemical, genetic, and microscopic analyses have revealed the stratified layout of FAs and identified the characteristic compendium of signalling and adaptor proteins, the so-called integrin adhesome [6–9]. While the essential roles of talin and kindlin in initiating integrin-based adhesion sites in various cell types have become clear [10], the function of other core adhesome proteins during the initial steps of FA formation is still debated.

For example, LIM (Lin-11, Isl1, MEC-3) domain containing adapter proteins are a highly enriched subgroup of integrin adhesome proteins thought to be involved in mechanosensing [7,11,12]. Individual LIM domains encompass approximately 60 amino acids forming a double zinc finger motif, which mediates binding to other proteins or nucleic acids [13,14]. A prominent member of this group of adapter proteins is paxillin, which contains 4 LIM domains and is ubiquitously expressed in mammalian tissues [15,16]. Paxillin is commonly employed as a marker protein of FAs and nascent focal complexes under various conditions, even where the normal morphology, function, and architecture of FAs is disturbed. Paxillin is one of the first proteins recruited to FAs [17] and efficiently localizes there even in the absence of myosin-generated forces [18]. Paxillin is found in nanometer distance from the plasma membrane with the carboxy-terminus detected in the same confined membrane-proximal layer as the cytoplasmic domain of integrin αv [9]. A main determinant of paxillin's efficient recruitment to integrins seems to be its association with kindlin2, which has been mapped to the amino-terminal LD domains and the carboxy-terminal LIM3 / LIM4 domain region of paxillin [19–21]. Recently, a crucial role of the paxillin LIM2 domain for FAs recruitment has been suggested [22], while initial work identified the paxillin LIM3 domain as being the essential determinant of its FA targeting [23]. A further mode of indirect recruitment of paxillin to integrin β tails can be mediated by the interaction of paxillin LD motifs with the talin R8 rod domain [24]. However, paxillin can localize to FAs in talin knockout (KO) cells [25], and paxillin is clearly incorporated into FAs in the absence of kindlins [26,27], suggesting additional currently uncharacterized, kindlin- and/or talin-independent mode (s) of paxillin's integrin engagement.

Here, we demonstrate that the paxillin LIM2 and LIM3 domains directly interact with carboxy-terminal residues of the integrin β subunit. Biochemical analysis of recombinant proteins, the NMR-based 3D structure of the paxillin LIM2/3 domains, and functional analysis of mutated paxillin and integrin β3 in vitro and in the cellular context reveal that this interaction is based on a flexible loop in paxillin's LIM3 domain, which presumably operates in a clamp-like mechanism to modulate cellular responses towards integrin ligands.

## Results

### Paxillin LIM2/3 domains can directly bind the cytoplasmic tails of integrin β1 and β3

In line with previous reports [11,18], we recently observed that paxillin can be recruited in the absence of mechanical forces to clusters of the integrin β CT, similar to the behaviour of known integrin binding partners such as talin and kindlin2 [28]. We wondered whether recruitment to clustered integrin β tails is a general feature of LIM domain containing adhesome proteins. To this end, we used CEACAM-integrin β CT fusion proteins (CEA-ITGB1 or CEA-ITGB3), which can be engaged from the outside of the cell by multivalent CEACAM binding bacteria (*Neisseria gonorrhoeae*). This process initiates microscale accumulation of free integrin β tails mimicking nascent adhesion formation and was therefore named Opa protein triggered integrin clustering (OPTIC) (S1A Fig) [28]. Interestingly, when coexpressed with CEA3-ITGB1 or CEA3-ITGB3, only paxillin and the closely related proteins Hic-5 and leupaxin showed a significant enrichment (Figs 1A, S1B and S1C), while other LIM domain

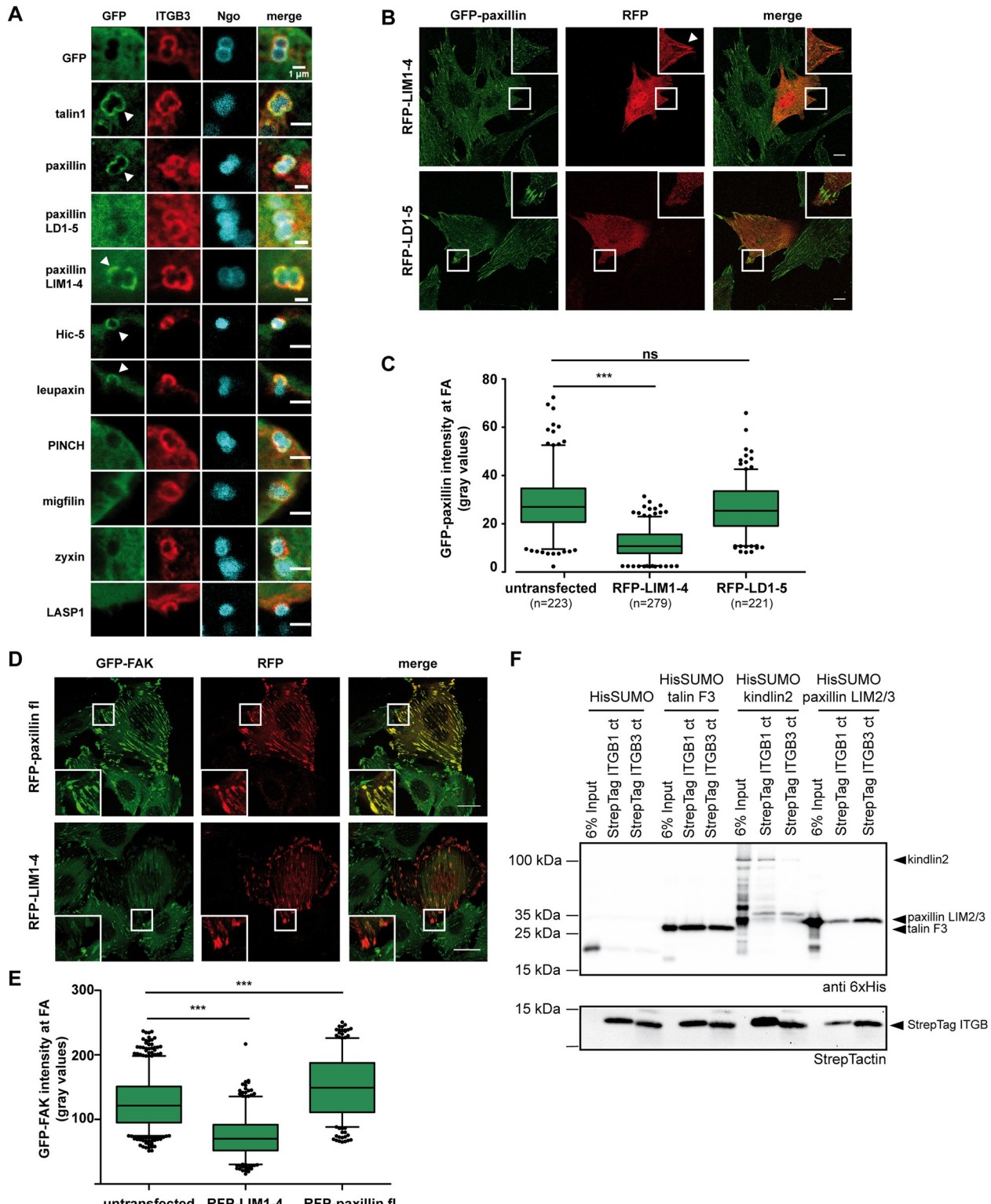

**Fig 1. The LIM2/3 domains direct paxillin to the cytoplasmic tail of integrin β1 and β3.** (**A**) 293T cells were transiently cotransfected with a CEACAM3-ITGB3 (ITGB3) fusion construct together with GFP or the indicated GFP-labelled proteins. Cells were seeded on poly-L-lysine and incubated for 1 h with Pacific Blue–labelled *Neisseria gonorrhoeae* (Ngo; blue) to cluster ITGB3 (stained red with α-CEACAM antibody). Recruitment of GFP-fused proteins to the clustered integrin β3 tail is indicated by white arrowheads. Bars represent 2 μm. (**B**) Stable GFP-paxillin expressing Flp-In 3T3 cells were transiently transfected with RFP-tagged paxillin LIM1-4 or RFP-tagged paxillin LD1-5 domains. Displacement of GFP-paxillin by

RFP-LIM1-4, but not RFP-LD1-5, is visible in boxed and enlarged areas. (**C**) GFP-Paxillin localization at FAs in presence of LIM1-4 or LD1-5 domains as in (B) was evaluated by measuring the GFP fluorescence intensity. Shown are mean values of GFP-paxillin intensity from 3 independent experiments. Total number of analysed FAs is given in brackets under each sample. Error bars represent 5 and 95 percentiles. Significance was calculated using one-way ANOVA, followed by Bonferroni multiple comparison test (*** $p < 0.0001$, ns = not significant). The data underlying this panel can be found in S1 Data. (**D**) GFP-FAK expressing MEFs were transiently transfected with RFP-paxillin full-length or RFP-LIM1-4. Displacement of GFP-FAK by RFP-LIM1-4, but not RFP-Paxillin, is visible in boxed and enlarged areas. (**E**) Calculation of GFP-FAK intensity at FAs was performed as in (C) $n = 360$ FAs per sample from 3 independent experiments. Significance was calculated using one-way ANOVA, followed by Bonferroni multiple comparison test (*** $p < 0.0001$). The data underlying this panel can be found in S1 Data. (**F**) In vitro pulldown using recombinant Strep-tag integrin β1 or β3 cytoplasmic tails and recombinant talin1 F3 domain, full-length kindlin2 or paxillin LIM2/3 domain fused to His-SUMO, or His-SUMO only as negative control.

proteins did not accumulate at clustered integrin β tails (Figs 1A, S1C and S1D). Paxillin recruitment to integrin clusters was dependent on the LIM domains, since the paxillin C-terminus encompassing LIM1-LIM4, but not the isolated N-terminal LD1-LD5 domains, strongly accumulated at integrin β cytoplasmic tails (Fig 1B and 1D), and paxillin LIM1-LIM4, but not the LD1-LD5 domains, displaced full-length paxillin together with its binding partner FAK from FA sites (Fig 1C–1E). These results indicated that paxillin, leupaxin, and Hic-5 differ from other LIM domain containing adhesome proteins by their ability to locate at clustered integrin β tails. Moreover, pulldown assays with purified, recombinant proteins demonstrated that, similar to the talin F3 domain and Kindlin2, the paxillin LIM2/3 domains can bind the cytoplasmic tails of integrin β1 and β3 in the absence of other cellular proteins (Fig 1F). These findings suggested an important role of the LIM2/3 domains to directly connect paxillin with integrin β tails.

## The solution structure of paxillin LIM2/3 reveals a flexible loop region in the LIM3 domain involved in binding to the integrin β3 carboxy terminus

To investigate the direct interaction between paxillin and the integrin β subunit in more detail, we used NMR spectroscopy to gain structural insight and to delineate the binding interface. Since LIM3 and, to a lesser extent, LIM2 have been shown to be mainly responsible for FA targeting of paxillin [23], we expressed the LIM2/3 tandem domain (aa380-499) of human paxillin and determined its solution structure based on heteronuclear multidimensional NMR experiments [29] (Fig 2A). Not surprisingly, regarding their high sequence similarity (39% identity when aligning residues $P_{381}$-$D_{436}$ of LIM2 with $P_{440}$-$E_{495}$ of LIM3; S2B Fig), both the LIM2 and the LIM3 domain of paxillin exhibit the characteristic double zinc finger motif as described for other LIM domain-containing proteins and paxillin family members [13,14,30,31] (Fig 2B and 2C). Each domain comprises 2 orthogonally packed β-hairpins, followed by an α-helix (Fig 2B and 2C). Interestingly, the LIM2 and LIM3 domains are connected by a short linker of 4 amino acids (F438-K441). There are in total only 19 long-range NOEs connecting 2 residues of the LIM2 domain (K432 and F435) with 4 residues of the linker region and the LIM3 domain, respectively (K440, R446, T458, and L459).

According to the measurements of the heteronuclear NOE between $H^N$ and N of the amides, the linker between LIM2 and LIM3 shows only slightly higher flexibility on the ps-to-ns timescale than the domains themselves (S2A Fig). However, the reduction in long-range NOE contacts in this region suggests that the structure is less densely packed here, resulting in differential relative orientations of the domains (Fig 2B and 2C). This would fit into the previously proposed scenario of the LIM domains as a sort of molecular ruler and/or tension sensor [11].

The heteronuclear NOE discloses further regions of increased flexibility in both domains, namely, around K393, and E419 in the LIM2 domain, and E451, and S479 in LIM3. All these regions are located in loops leading from the first 2 ligands in each zinc finger to the following β-hairpins in each LIM domain, respectively.

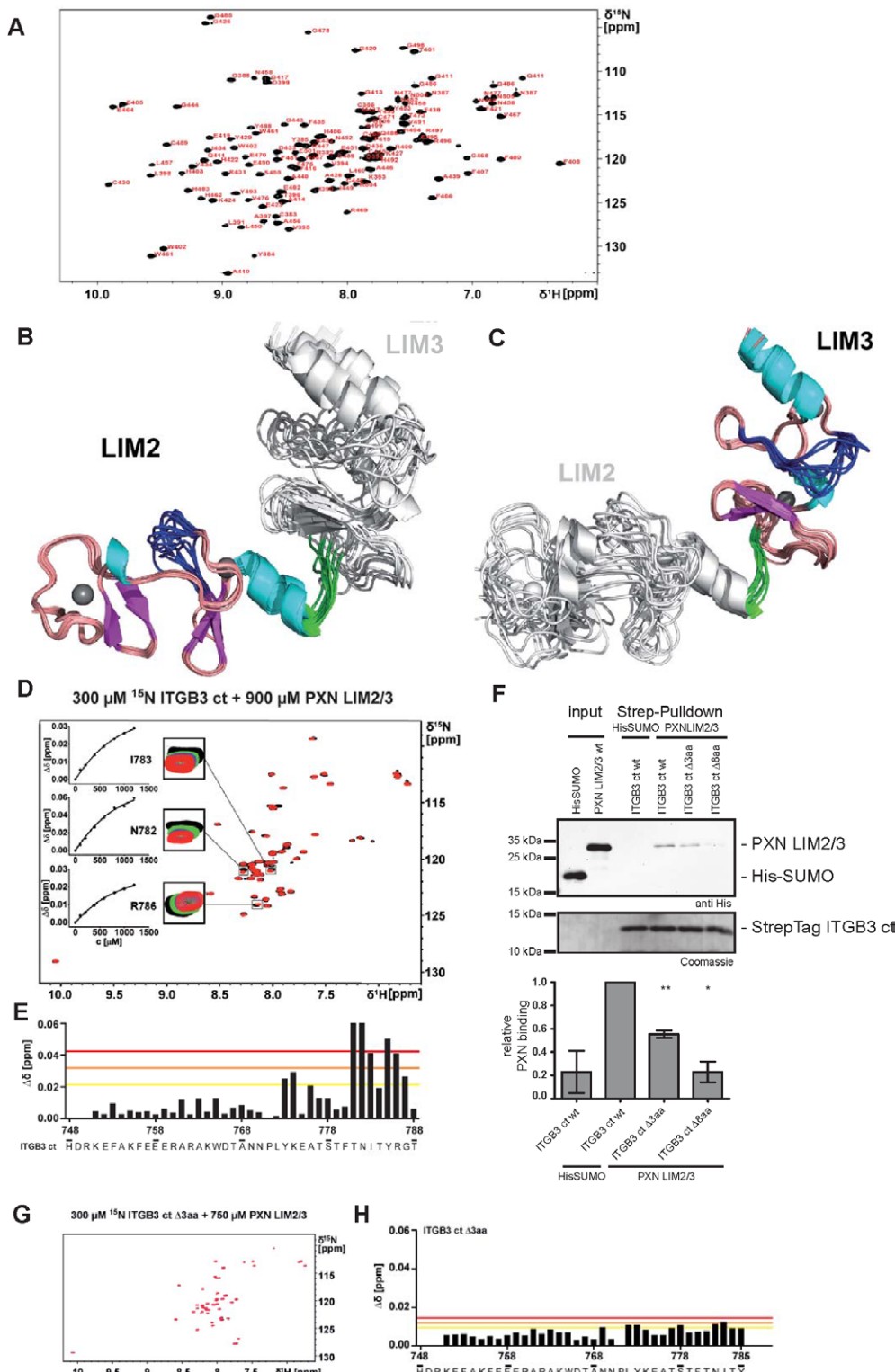

**Fig 2. The paxillin LIM2/3 domains directly interact with C-terminal residues of integrin β3.** (**A**) Assigned $^{1}$H-$^{15}$N-HSQC spectrum of paxillin LIM2/3. Backbone amide crosspeaks and side-chain amide groups are labeled by amino acid type and position. (**B**, **C**) Solution structure of paxillin LIM2/3. The final ensemble of 10 conformers with lowest target function is shown fitted to the LIM2 domain (residues P381 to F438) (**B**) or fitted to the LIM3-domain (residues P440 to R497) (**C**), shown in ribbon representations. In the fitted part, α-helices are colored cyan and β-strands

magenta. Zinc ions are shown as grey spheres. The linker region between LIM2 and LIM3, residues F438-A439-P440-K441, is colored in green. The flexible loops of the LIM2 domain (residue F415 to F421) and the LIM3 domain (residue T473 to E482) are shown in blue. The domain that was not used for fitting is shown in light grey. (**D**) $^{15}$N-HSQC titration of 300 μM $^{15}$N integrin β3 ct (ITGB3 ct) with paxillin LIM2/3. Paxillin was added in concentrations up to 900 μM. Boxes show a selection of signals affected by CSPs (residues I783, N782, and R786) in the presence of 0 μM (black), 150 μM (green), 300 μM (blue), and 900 μM (red) paxillin LIM2/3. Insets show the concentration dependence of combined amide CSPs globally fitted to a one site binding model. (**E**) Combined amide CSPs of 300 μM $^{15}$N integrin β3 ct in the presence of 760 μM paxillin LIM2/3 vs. residue number of integrin β3 ct. Lines indicate average $\delta\Delta$ + 1× s.d. (yellow), $\delta\Delta$ + 2× s.d. (orange), and $\delta\Delta$ + 3× s.d. (red). (**F**) In vitro pulldown of His-SUMO or His-SUMO-paxillin LIM2/3 (PXN LIM2/3) using the Strep-tag integrin β3 cytoplasmic tail in the wild-type form (wt) or with a truncation of the carboxy-terminal 3 (Δ3aa) or 8 (Δ8aa) amino acids. The bar graph below shows the densitometric quantification of the pulldown experiments ($n = 3$). Statistical significance was calculated using one-sample $t$ test to calculate if samples mean are significantly different from a hypothetical value of 1 (* $p < 0.05$, ** $p < 0.01$). The data underlying this panel can be found in S1 Data. (**G**) $^{15}$N-HSQC titration of 300 μM $^{15}$N integrin β3 Δ3aa (ITGB3 Δ3aa) with paxillin LIM2/3 (PXN LIM2/3). Paxillin was added up to a concentration of 750 μM. (**H**) Combined amide CSPs of 300 μM $^{15}$N integrin β3 ct Δ3aa in the presence of 750 μM paxillin LIM2/3 vs. residue number of integrin β3 ct Δ3aa.

Interestingly, by analyzing the heteronuclear NOEs and the structural definition of the final ensemble of conformers, we identified a 7-amino acid stretch (F475-F481) in the second zinc finger of the LIM3 domain that constitutes a flexible, surface exposed loop in the free protein (Fig 2C). This loop is flanked by F475 and F481 and is situated adjacent to a hydrophobic patch or groove. In this region, the amino acid sequence and, in particular, residues F475, F480, and F481 are highly conserved within paxillin family members and across species (S2B Fig). We speculated that this flexible loop might act in concert with its opposing residues (E451, N452, and Y453) of the first zinc finger and the hydrophobic patch to support a clamp-like mechanism for integrin β CT binding. Noteworthy, the LIM2 domain harbors a similarly flexible loop, yet with a negatively charged residue at its center (E419) and a less hydrophobic patch opposing it (LIM2: K$_{393}$VVTALD$_{399}$ versus LIM3: N$_{452}$YISALN$_{458}$; S2B Fig).

To identify the binding region of paxillin LIM2/3 on the integrin β CT, we titrated unlabeled paxillin LIM2/3 to $^{15}$N-labeled cytoplasmic tails of human integrin β1 (aa758-798) or β3 (aa748-788), respectively. In both cases, significant chemical shift perturbations (CSPs) of specific integrin residues were observed (Figs 2D, 2E, S3A and S3B). A dissociation constant (K$_D$) of 52 ± 30 μM was apparent for integrin β1 (S3A Fig), while integrin β3 showed a higher K$_D$ (528 ± 130 μM) (Fig 2D). As expected, regarding the higher K$_D$ value in case of integrin β3, the titration curves do not reach full saturation but end at ca. 70% fraction bound. This is limited by the maximum concentration of unlabelled binding partner that could be reached and limits, in turn, the accuracy of the K$_D$ value determined.

These findings are in line with our previous microscopic observations, where recruitment of paxillin to the clustered integrin β1 cytoplasmic tail was more pronounced (S1C and S1D Fig). Surprisingly, when mapping the CSPs onto the primary sequence of integrin β CTs, the interacting regions were distinct. While the highest CSPs in integrin β1 CT were distributed over the membrane proximal NPxY motif and a neighbouring conserved TT motif (S3B Fig), the largest chemical shifts in integrin β3 were confined to the 8 C-terminal amino acids, spanning the membrane distal NxxY motif (Fig 2E). Indeed, deleting 8 amino acids from the C-terminus of integrin β3 (Δ8aa) completely abrogated paxillin LIM2/3 binding in pulldown experiments, while deletion of the last 3 amino acids (Δ3aa) significantly reduced paxillin binding (Fig 2F). Furthermore, applying integrin β3 CTs Δ3aa or Δ8aa in titration experiments with paxillin LIM2/3 yielded no significant CSPs, confirming the loss of interaction (Figs 2G, S3C and S3D). The biochemical data were further corroborated by OPTIC assays, were both the ITGB3 Δ3 and the Δ8 mutant, but not mutation of the kindlin binding site (integrin β3 S778A) [32,33], diminished paxillin recruitment to clustered integrin β3 tails (S3E and S3F Fig). As expected, association of the talin F3 domain with the integrin β3 tail was not

compromised by C-terminal deletion of 3 or 8 amino acids (S3G Fig), supporting the idea that the direct binding of paxillin to the carboxy-terminal residues of integrin β3 could be relevant in a cellular context.

## The flexible loop in paxillin LIM3 mediates a direct association with the integrin β3 cytoplasmic tail

To precisely identify the integrin binding site within the paxillin LIM2/3 domains, we titrated the unlabeled CT of human integrin β3 to $^{15}$N-labelled paxillin LIM2/3. Similar to the inverse titrations, the $K_D$ value of this interaction was determined to 532 ± 239 μM (Fig 3A). It should be noted here that the integrin β3 peptide can be added to higher final concentrations compared to the titration of $^{15}$N-labeled integrin β3 with unlabelled paxillin (Fig 2D). Therefore, the maximum CSPs observed in Fig 3A are larger than those in Fig 2D. Changes in the arrangement of aromatic sidechains with respect to the amide groups of the flexible loop may play an additional role. Importantly, the most prominent CSPs in paxillin were recorded within the second zinc finger of the LIM3 domain, specifically in the flexible loop region between F475 and F481 (Fig 3B). To verify our NMR-based epitope mapping, we individually mutated residues of the loop region. Interestingly, exchanging phenylalanine F475, F480, or F481 for alanine caused a complete or partial unfolding of the LIM3 domain (S4A–S4C Fig). Although these residues show strong CSPs and participate in integrin binding, these phenylalanines are also essential for maintaining the structure of the LIM3 domain. To gain further insight into the role of the loop region, these phenylalanines were left intact, but instead, the residues between F475 and F480 were mutated individually (V476A, S479A) or in combination (V476-S479) to alanine (LIM2/3-4A). Individual point mutations as well as Paxillin LIM2/3-4A exhibited a stable LIM domain fold (S4D-S4F Fig). However, when titrating $^{15}$N-labelled LIM2/3-4A with up to millimolar concentrations of integrin β3, no saturable CSPs could be detected (Fig 3C and 3D). The relatively large CSPs of a few isolated residues (F421, Y453, and F480) did not saturate and, therefore, do not indicate specific binding of the 4A-variant towards ITGB3. Instead, these residues seem to be quite sensitive to slight changes in buffer composition. Although identical buffer solutions during NMR sample preparation were used, such differences may occur during up-concentration or as a consequence of variations within the purification protocols of paxillin-LIM2/3 versus the integrin peptides. Pulldown assays with the paxillin LIM2/3-4A mutant confirmed diminished binding to the integrin β3 CT (Fig 3E). Next, we introduced the LIM3 loop mutations into full-length paxillin (GFP-PXN-4A) to corrupt the direct engagement of integrin β3 in intact cells. In addition, we generated a truncated paxillin lacking the LIM4 domain (GFP-PXN ΔLIM4) to interfere with the kindlin-mediated indirect binding of paxillin to the integrin β subunit as reported by Zhu and colleagues [19]. GFP-tagged paxillin (PXN) wild type (wt), GFP-PXN-4A, or GFP-PXN ΔLIM4 were stably reintroduced into paxillin KO fibroblasts (S5A Fig). When cell spreading on the integrin ligands vitronectin and fibronectin was monitored, spreading of paxillin KO cells was strongly impaired and reexpression of GFP-paxillin wt rescued this phenotype (Figs 3F and S5B). In contrast, the GFP-PXN-4A mutant was not able to revert the spreading defect and mimicked paxillin KO cells in the first 2 h after seeding on the substrate, while paxillin lacking the LIM4 domain reconstituted cell spreading partially (Figs 3F and S5B). PXN wt and PXN ΔLIM4 also reverted the round, circular morphology of the paxillin-KO fibroblasts to the spindle-shaped, pointed cell phenotype of wt fibroblasts, whereas cells reexpressing GFP-PXN-4A retained the elevated circularity of the paxillin KO cells (Figs 3F and S5B). On vitronectin, both GFP-PXN-4A and GFP-PXN ΔLIM4 showed slightly reduced presence at FA sites (S5C and S5D Fig), but only in GFP-PXN-4A cells a reduction in overall FA area was

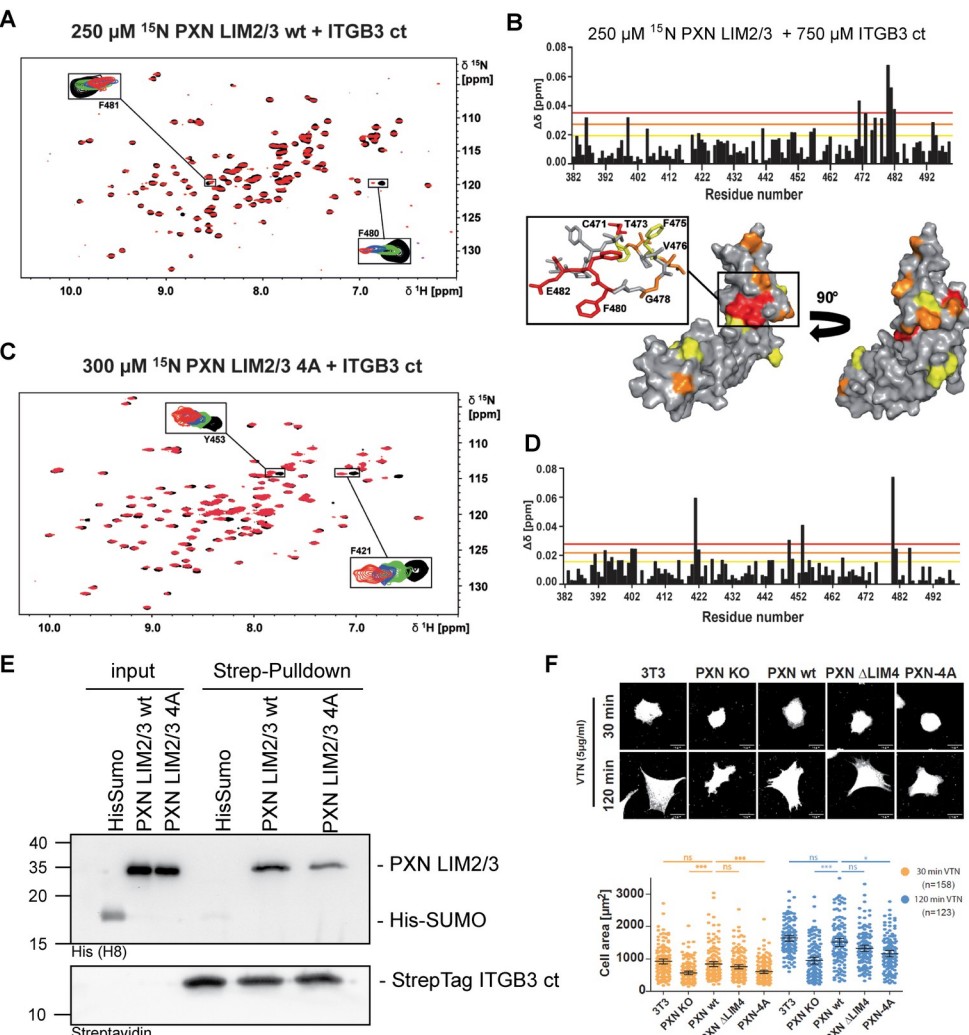

**Fig 3. A flexible loop in the LIM3 domain mediates binding to the integrin β cytoplasmic tail. (A)** $^{15}$N-HSQC titration of 250 μM $^{15}$N paxillin LIM2/3 wt (PXN LIM2/3 wt) with integrin β3. ITGB3 ct was added in concentrations up to 2,420 μM. Boxes show a selection of signals affected by CSPs (residues F480 and F481) in the presence of 0 μM (black), 200 μM (green), 600 μM (blue), and 2420 μM (red) integrin β3 ct. **(B)** Combined amide CSPs of 250 μM $^{15}$N paxillin LIM2/3 in the presence of 750 μM integrin β3 vs. residue number of paxillin. Lines indicate average δΔ + 1× s. d. (yellow), δΔ + 2× s.d. (orange), and δΔ + 3× s.d. (red). The amide CSPs were mapped onto the solution structure of paxillin LIM2/3 shown as surface representation from 2 perspectives. Residues showing CSPs larger than average δΔ + 3× s.d. are colored red, residues for which [average δΔ + 3× s.d. < δΔ < average δΔ +2× s.d.] are colored orange, and residues for which [average δΔ + 2× s.d.< δΔ < average δΔ + 1× s.d.] are colored yellow. Residues with δΔ < average + 1× s.d. are colored grey. The boxed region is also shown in stick representation, including the flexible loop of the LIM3 domain using the same color code. Residues experiencing significant CSPs are labelled by amino acid type and position. **(C)** $^{15}$N-HSQC titration of 300 μM $^{15}$N paxillin LIM2/3 4A (PXN LIM2/3 4A) with integrin β3. Integrin was added in concentrations up to 1582 μM. Boxes show a selection of signals affected by CSPs (residues F421 and Y453) in the presence of 0 μM (black), 632 μM (green), 1,107 μM (blue), and 1,582 μM (red) PXN LIM2/3 4A. **(D)** Combined amide CSPs of 250 μM $^{15}$N PXN LIM2/3 4A in the presence of 630 μM integrin β3 vs. residue number of paxillin. Lines indicate average Δδ+ 1× s.d. (yellow), Δδ + 2× s.d. (orange), and Δδ + 3× s.d. (red). **(E)** In vitro pulldown of recombinant His-SUMO PXN LIM2/3 wt or His-SUMO PXN LIM2/3 4A using Strep-Tag ITGB3 ct wt. PXN LIM2/3 4A shows reduced binding to ITGB3. **(F)** Spreading of paxillin KO cells, stably reexpressing empty vector (PXN KO), GFP-paxillin wt (PXN wt), or paxillin mutants GFP-PXN ΔLIM4 or GFP-PXN-4A. Starved cells were seeded for 30 or 120 min on the integrin β3 ligand vitronectin. The membrane of fixed cells was stained with CellMask Orange to visualize total cell extension. Scale bars represent 20 μm. Below is the quantification of stained area/individual cell at 30 min (orange) and 120 min (blue). Sample sizes are given in brackets. Statistical significance was calculated using one-way ANOVA followed by Bonferroni multiple comparison test (ns: not significant; *** $p \leq 0.001$; ** $p \leq 0.01$). The data underlying this panel can be found in S1 Data.

apparent (S5C and S5E Fig). Furthermore, in paxillin KO cells as well as in GFP-PXN ΔLIM4 and GFP-PXN-4A cells, the presence of kindlin at FA sites appeared diminished (S5C Fig). These results indicate that paxillin and, in particular, the direct interaction between the flexible loop of the paxillin LIM3 domain and integrin β3 is of importance for FA composition and function.

### Paxillin directly associates with integrin β3 in the absence of kindlins to promote cell spreading

To finally corroborate the physiological relevance of the direct interaction between integrin β3 and paxillin LIM domains, we first generated integrin β3 KO fibroblasts via CRISPR/Cas9 and stably reintroduced either full-length integrin β3 wt or the truncated integrin β3 variants, Δ8aa or Δ3aa, which lack the newly characterised paxillin binding site (Fig 4A). Expression of other FA proteins was not altered in these cells (Fig 4B). While integrin β3-deficient cells exhibited a strongly impaired initial spreading on vitronectin- and fibronectin-coated substrates, the reexpression of integrin β3 wt reverted this phenotype (Figs 4C and S5F). In contrast, integrin β3 Δ3aa and, even more so, integrin β3 Δ8aa reexpressing fibroblasts were impaired in their spreading ability (Figs 4C and S5F). A similar spreading defect on integrin ligands has also been reported for kindlin-deficient cells, and deletions of the integrin carboxy terminus might also corrupt the kindlin binding site, indirectly affecting the recruitment of paxillin. To substantiate our biochemical findings of a direct paxillin interaction with the integrin β subunit, we employed kindlin1- and kindlin2-deficient double KO cells [20,26]. As these cells display strongly diminished expression of integrin β3 compared to wt fibroblasts, we introduced integrin β3 wt, integrin β3 Δ8aa, or integrin β3 Δ3aa into the kindlin1/2 KO cells (Fig 4D). In the kindlin KO cells, the reexpression of integrin β3 wt or its variants did also correct the reduced levels of integrin αv but did not affect levels of other FA proteins (Fig 4D and 4E). In line with the lack of integrin β3 expression, the mock-transfected kindlin1/2-KO fibroblasts hardly attached and did not spread on vitronectin-coated substrates (Fig 4F). This phenotype has been reported before and has been used to demonstrate the essential role of kindlins in integrin-mediated cell adhesion [20]. However, even in the absence of kindlin1/2, cell spreading on vitronectin was partially recovered upon expression of integrin β3 wt (Fig 4F). Importantly, the newly appearing adhesion sites in integrin β3 wt expressing cells stained positive for paxillin and expression of integrin β3 mutants with a truncated paxillin binding site did not support cell attachment on vitronectin (Fig 4F). Furthermore, paxillin-positive adhesions in integrin β3 wt expressing kindlin1/2-KO cells appeared upon plating onto vitronectin, but not upon plating onto poly-L-lysine, and these integrin β3-mediated contacts also stained positive for talin (Fig 4F). Altogether, our results provide evidence of a direct binding interaction between the distal carboxy terminus of integrin β3 and a flexible loop in the paxillin LIM3 domain. This direct association localizes paxillin to FAs in the absence of kindlins and modulates integrin β3-initiated cellular responses.

## Discussion

Although paxillin was discovered more than 30 years ago and constitutes a core FA protein, its mode of FA targeting has remained controversial. Recent biochemical approaches have pointed to an indirect association of paxillin with the integrin β1 and β3 subunits via the integrin binding partner kindlin1 or kindlin2 [21,26]. As the association of kindlin with paxillin appears to rest on the paxillin LD repeats and the LIM4 domain [19,20], these findings do not explain the central role of the LIM3 domain for FA localization, which has been delineated by microscopic observations in intact cells [23]. Here, we present evidence for a direct interaction

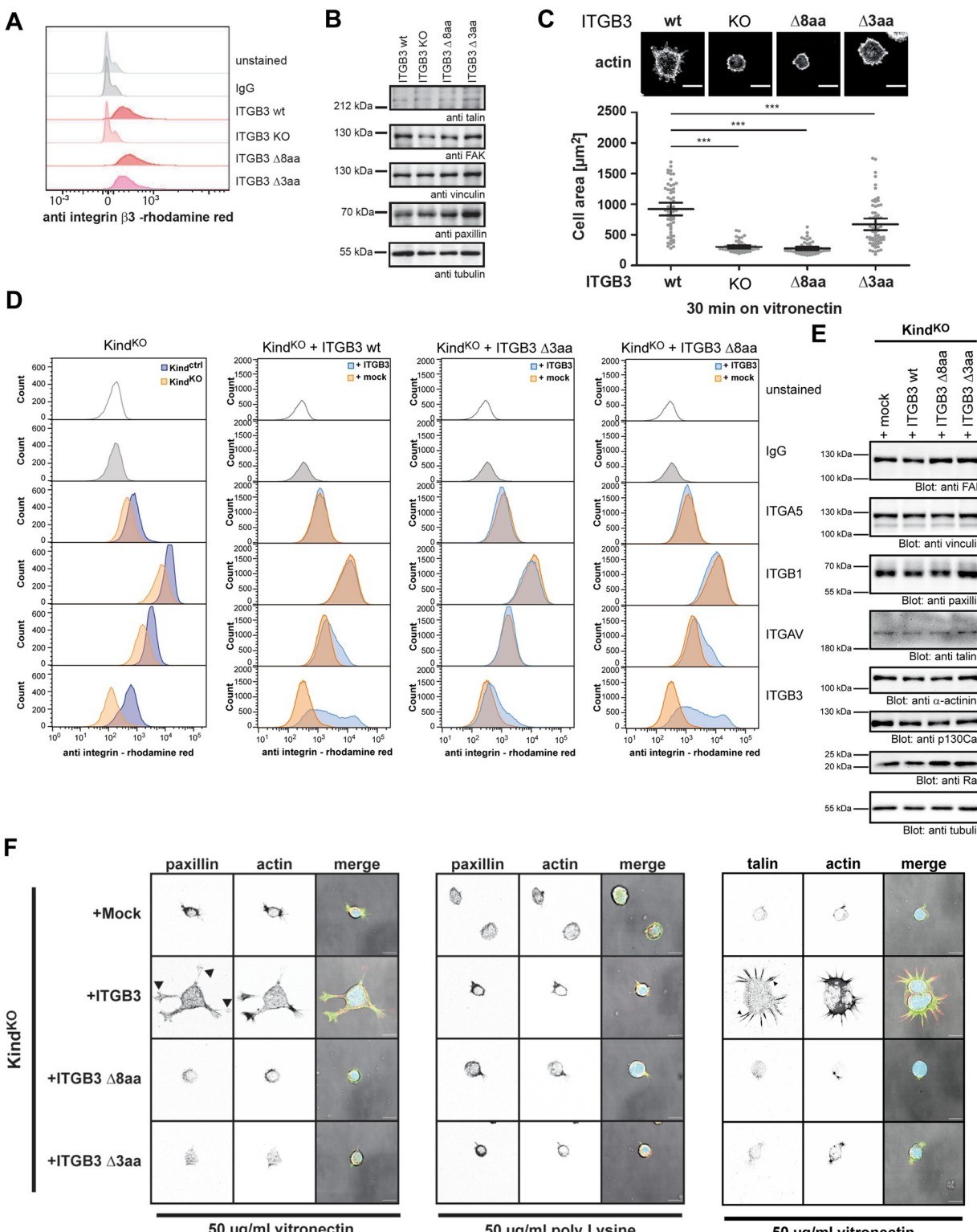

**Fig 4. The paxillin–integrin β3 interaction allows cell attachment in the absence of kindlins. (A)** Flow cytometric analysis of NIH Flp-In integrin β3 KO and the indicated ITGB3 wt, ITGB3 Δ8aa, or ITGB3 Δ3aa reexpressing cell lines. Cells were left unstained or stained against mouse integrin β3 or with an isotype matched control IgG. (**B**) WCLs of cell lines in (A) were probed with the indicated antibodies directed against core FA proteins. Tubulin was used as loading control. (**C**) Serum starved cells as in (A) were seeded onto vitronectin-coated (5 μg/ml) glass slides for 30 min, and cell area was measured. Shown are mean values and 95% confidence intervals of *n* = 60 cells per sample from 3

independent experiments. Statistical significance was calculated using one-way ANOVA followed by Bonferroni multiple comparison test (*** $p \leq 0.001$). The data underlying this panel can be found in S1 Data. (**D**) Flow cytometric analysis of Kindlin 1/2 flox cells (Kind^Ctrl), Kindlin 1/2 KO (Kind^KO), or Kind^KO cells stably transduced with either murine full-length integrin β3, truncated integrin β3 Δ8aa or Δ3aa, or empty vector backbone (mock). Cells were analysed for their surface expression of various integrin subunits using flow cytometry. (**E**) WCL from cell lines in (D) were analysed by western blotting with antibodies against indicated core FA proteins. Monoclonal α-tubulin antibody was used as loading control. (**F**) Serum starved cells as in (D) were seeded on glass coverslips coated with 50 μg/ml vitronectin or poly-Lysine for 4 h. Cells were fixed and stained for endogenous paxillin or talin as indicated. Paxillin- and talin-positive cell attachment sites in ITGB3 wt expressing kindlin1/2 KO cells are indicated with black arrowheads.

between the paxillin LIM3 domain and the cytoplasmic tails of integrin β1 and integrin β3, respectively. Together with the indirect link provided by kindlin, the intimate association of paxillin with the integrin β subunit now unveils the full spectrum of paxillin's FA recruitment modalities and unmasks the fundamental building principles of cellular attachment sites.

In solution, paxillin's LIM2 and LIM3 domains adopt an overall fold consistent with available structural data for LIM domains of other proteins [13,14,30,31]. However, our NMR structure reveals an intriguing detail, which is conserved in paxillin orthologues from other species. Indeed, the LIM3 domain of paxillin harbours a flexible, surface-exposed loop, which, based on sequence homology, is also present in the paxillin family members Hic-5 and leupaxin. This loop demarcates the integrin binding site in the LIM3 domain and appears to function as a clasp to stabilize the association of paxillin with the integrin cytoplasmic tail. Noteworthy, LIM2 and LIM3 share significant sequence similarity (S2B Fig). This, together with their tandem arrangement, suggests that these LIM domains originate from a gene duplication event as has been proposed by others on a broader context of LIM domain-containing proteins [34].

This additional direct interaction between the integrin β cytoplasmic tail and paxillin's LIM3 domain now consolidates an emerging principle of FA organization: Each core component of FAs, including talin, kindlin, paxillin, FAK, vinculin, and α-actinin, is able to sustain multiple, independent interactions with other FA components. In analogy to a steel frame construction, this kind of assembly not only allows a stepwise expansion of the protein complex but also provides a further mechanical reinforcement with every incoming component. In the specific example of paxillin, this protein can associate via its LIM4 domain with integrin-bound kindlin2 [19], but then paxillin will also be in place to utilize the flexible loop in its LIM3 domain to bind the integrin β subunit and to reenforce this tripartite complex as a prerequisite for efficient initial cell spreading. Furthermore, as paxillin can interact via its LIM1/2 domains with the talin head [22] and via its LD1 domain with the talin rod region [24], there is the possibility that paxillin strengthens the early integrin-associated protein complex beyond kindlin. Indeed, a stabilization of the integrin-talin-kindlin nexus by paxillin has been observed [21]. On first sight, such a cooperative binding scenario seems implausible as the binding sites of paxillin and kindlin at the carboxy terminus of the integrin β3 subunit partially overlap. However, a recent study has highlighted an unforeseen behaviour of asymmetric affinity regulation by integrin tail allostery, where kindlin binding increases the affinity of talin to associate with the same integrin tail, whereas the binding of talin, in turn, decreases kindlin's affinity for its neighbouring binding site [35]. This surprising finding suggests that while kindlin exhibits cooperative binding behaviour towards talin-association with integrin β1, talin shows competitive behaviour towards kindlin-binding to the same integrin tail. However, despite the decreased kindlin–integrin affinity in the presence of talin, a ternary complex between integrin, talin, and kindlin is observed, as this ternary complex is stabilized by additional talin–kindlin interactions [35].

We also can envision that the flexible loop of paxillin LIM3 enables this protein to associate with a minimal binding site encompassing the carboxy-terminal amino acids of the integrin β3

tail, even though this binding site partially overlaps with the kindlin binding site and vice versa. However, such a presumably competitive behaviour with regard to direct integrin association might be compensated by the binding interactions between paxillin and kindlin/talin, which are mediated by paxillin's LIM4 and LD domains, respectively. While the affinity of an individual interaction between paxillin and integrin β3 with $K_D$ approximately 500 μM is surprisingly weak, many of such interactions are formed during FA assembly. Similar to other well-known, multivalent interactions, such as those mediated by oligosaccharide–lectin interactions in cell–cell recognition, such interactions may become—through avidity—very strong but still allow for fast assembly and disassembly through Velcro-like mechanisms. In this regard, multiple reciprocal interactions between talin, kindlin, paxillin, and the integrin β tail would also increase the avidity and might demonstrate once more how multivalent low-affinity interactions such as those observed for paxillin LIM3 and integrin β3 stabilize macromolecular networks. Accordingly, the relatively low affinity between LIM3 and integrin β3 might be a prerequisite for efficient assembly and disassembly of FAs.

Intriguingly, such reciprocal interactions between FA core components could also be the basis for the astonishing flexibility in the temporal sequence, in which these proteins can assemble at integrin cytoplasmic domains. For example, talin binding to the integrin β tail appears as a prerequisite for the recruitment of its binding partner FAK [36,37], while the opposite sequence of assembly has also been reported [38,39]. This behaviour is mirrored by kindlin and paxillin, as kindlin is able to recruit paxillin, while the direct binding of paxillin to the β subunit can turn this sequence of events on its head and could potentially allow paxillin-dependent recruitment of kindlin. Interestingly, one of the factors, which determines the order of assembly, appears to be the nature of the involved integrin heterodimer. In particular, differences between integrin α5β1 and integrin αvβ3 do not only exist with regard to the ligand spectrum but also how they convey the ligand binding event into the cell [10]. In this regard, α5β1 integrins are known to determine adhesion strength and form catch bonds with their ligands, when increasing forces are applied [40,41]. The elevated binding affinity of paxillin for integrin β1 and its ability to associate with talin and kindlin might reflect this need to withstand high forces. Although integrin αvβ3 is not able to sustain the high binding strength of α5β1 integrin, integrin αvβ3 exhibits faster binding rates and stimulates integrin α5β1-mediated binding [42,43]. Interestingly, the fast binding rate of integrin αvβ3 correlates with its increased ability to recruit paxillin and its propensity to initiate larger paxillin-positive adhesion sites [43,44]. Furthermore, on patterned substrates, cell spreading and paxillin recruitment preferentially occur via the vitronectin-binding integrin αvβ3 [45]. Together with our findings of a prominent recruitment of paxillin to integrin β3 in kindlin-deficient cells and of reduced spreading of paxillin LIM3-4A expressing cells on vitronectin, all these observations suggest a particularly prominent role for the direct association of the paxillin LIM3 domain with integrin β3. It is also interesting to note that the paxillin LIM3 domain appears to latch onto a specific section of the integrin β3 subunit at the far carboxy terminus, which shows significant CSPs upon binding. The C-terminal amino acids of integrin β3 differ from all other β subunits, making this a unique recognition site and helping to explain this peculiar interaction mode of paxillin and integrin β3. These residues can also serve as an unconventional binding site for the SH3 domains of various Src family kinases, adding to Src kinase activation upon integrin engagement [46–48]. Accordingly, a reduction in Src kinase activity could also contribute to the spreading phenotype of kindlin KO cells expressing truncated integrin β3 variants.

Although further studies are needed to delineate the stoichiometry of talin, kindlin, and paxillin at clustered integrin β subunits, our structural elucidation of the paxillin LIM2 and LIM3 domains and their association with the integrin β carboxy terminus now provides the

foundation to probe and manipulate the functional contribution of paxillin to matrix adhesion and cell spreading.

## Materials and methods

### Antibodies and dyes

The following primary and secondary antibodies were used at indicated concentrations: anti-human α-actinin1 (mouse monoclonal, BM75.2, Sigma Aldrich, A5044; WB 1:1,000), anti-human talin (mouse monoclonal, 8d4, Sigma Aldrich, T3287; WB 1:750), anti-human FAK (rabbit polyclonal, A-17, Santa Cruz, sc-557; WB 1:250), anti-human kindlin2 (mouse monoclonal, 3A3, Merck, MAB2617; WB 1:1,000, IF 1:200), anti-mouse kindlin2 (rabbit polyclonal, 11453-1-AP, Proteintech; WB 1:2,000), anti-human cSRC (rabbit polyclonal, SRC2, Santa Cruz, sc-18; WB 1:1,000), anti-human ILK (rabbit monoclonal, EP1593Y, Epitomics; WB 1:1,000), anti-human p130Cas (rabbit polyclonal, N17, Santa Cruz; WB 1:1,000), anti-human vinculin (mouse monoclonal, VIN-1, Sigma Aldrich, V9131; WB 1:1,000), anti-human Hic-5 (mouse monoclonal, 34, BD Biosciences, 611164; WB 1:500), anti-human paxillin (mouse monoclonal, 5H11, Thermo Fisher Scientific, AHO0492; WB 1:1,000, IF 1:200), anti-GAPDH (mouse monoclonal, GA1R, Thermo Fisher Scientific, MA5-15738-HRP), anti-human Rac (rabbit polyclonal, invitrogen, PA5-17519; WB 1:1,000), anti-human CEACAM1, 3, 4, 5, 6 (mouse monoclonal, D14HD11, Aldevron; WB 1:6,000, IF 1:200); mouse monoclonal anti 6xHis (mouse monoclonal, HIS.H8, Thermo Fisher Scientific, MA1-21315; WB 1:2,000), anti GFP (mouse monoclonal, JL8, Clontech; WB: 1:6,000), anti-human tubulin (mouse monoclonal, E7, purified from hybridoma cell supernatants, Developmental Studies Hybridoma Bank, University of Iowa, USA; WB 1:1,000), anti-mouse integrin β1 (Armenian hamster monoclonal, Hmb1-1, Thermo Fisher Scientific, 11-0291-82; FC 1:300), anti-mouse integrin β3 (Armenian hamster monoclonal, 2C9.G3, Thermo Fisher Scientific, 13-0611-81; FC 1:200), anti-mouse integrin α5 (rat monoclonal, MFR5, BD Biosciences, 553319; FC 1:300), anti-mouse integrin αV (rat monoclonal, RMV-7, BD Biosciences, 550024; FC 1:300). Secondary antibodies used: horseradish peroxidase (HRP)-conjugated goat anti-mouse; WB 1:10,000, HRP-conjugated goat anti-rabbit; WB: 1:5,000; Cy5-conjugated goat anti-mouse; IF 1:200, Dylight 488 conjugated goat anti-mouse, IF 1:200, Rhodamine Red-X–conjugated goat anti-arm. Hamster; FC 1:300, Rhodamine Red–conjugated goat anti-rat; FC 1:300 (all from Jackson ImmunoResearch, Baltimore, USA). CellMask Orange Plasma membrane stain, Thermo Fisher Scientific, C10045; IF 1:1,000.

### Cell culture and transient transfection

Human embryonic kidney 293T cells (293T; American Type Culture Collection CRL-3216) were grown in DMEM supplemented with 10% calf serum. Flp-In 3T3 murine fibroblasts (Thermo Fisher Scientific) were cultured in DMEM supplemented with 10% fetal calf serum (FCS) and 1% nonessential amino acids. GFP-FAK expressing mouse embryonic fibroblasts (GFP-FAK MEFs) derived from FAK/p53 −/− KO MEFs [49] and Kindlin1/2 KO fibroblasts [20,26] were cultured in DMEM supplemented with 10% FCS and 1% nonessential amino acids on gelatine-coated (0.1% in PBS) cell culture dishes. All cells were maintained at 37°C, 5% $CO_2$, and subcultured every 2 to 3 days.

For transient transfection of 293T cells, cells were seeded at 25% confluence the day before and transfected using the standard calcium phosphate method with a total amount of 5 μg plasmid DNA/dish. For transient transfection of Flp-In 3T3 cells, $1 \times 10^5$ cells were seeded into 6-well plates the day before and transfected with using jetPRIME transfection reagent (Polyplus transfection, Illkirch, France), following manufacturer's protocol. GFP-FAK MEFs

were transiently transfected with Lipofectamin 2000, according to manufacturer's recommendations.

## Whole cell lysates (WCLs) and WB

WCLs were generated by lysing equal cell numbers in radioimmunoprecipitation assay buffer (1% Triton X-100, 50 mM Hepes, 150 mM NaCl, 10% glycerol, 1.5 mM $MgCl_2$, 1 mM EGTA, 0.1% wt/vol SDS, and 1% vol/vol deoxycholic acid) supplemented with freshly added protease and phosphatase inhibitors (10 mM sodium pyrophosphate, 100 mM NaF, 1 mM sodium orthovanadate, 5 μg/ml leupeptin, 10 μg/ml aprotinin, 10 μg/ml Pefabloc, 5 μg/ml pepstatin, and 10 μM benzamidine) and phosphatase saturating substrate (para-nitrophenolphosphate [pNPP]; Sigma-Aldrich; 10 mM). Chromosomal DNA was mechanically sheared by passing through a metal needle. DNA and cell debris were pelleted by addition of sepharose beads and centrifugation (13,000 rpm, 30 min, 4˚C). Supernatant was supplemented with 4× SDS sample buffer (4% wt/vol SDS, 20% wt/vol glycerol, 125 mM Tris-HCl, 20% vol/vol β-mercaptoethanol, and 1% wt/vol Bromophenol blue (pH 6.8)) and boiled for 5 min at 95˚C. Proteins were resolved on 10% to 18% SDS-PAGE. After separation, the proteins were transferred to a polyvinylidene fluoride membrane (Merck Millipore), followed by blocking in 2% BSA containing 50 mM Tris-HCl, 150 mM NaCl, and 0.05% Tween 20 (pH 7.5) (TBS-T) buffer. The membrane was incubated with primary antibody in blocking buffer overnight at 4˚C, washed 3 times with TBS-T, and incubated with HRP-conjugated secondary antibody in TBS-T for 1 h at RT. The chemiluminescent signal of each blot was detected with ECL substrate (Thermo Fisher Scientific) on the Chemidoc Touch Imaging System (Bio-Rad) in signal accumulation mode. Acquired images were processed in Adobe Photoshop CS4 by adjusting illumination levels of the whole image. Original uncropped and minimally adjusted images of all western blots can be found in S1 Raw Images.

## Recombinant DNA

The construction of $His_6SUMO$-tagged talin F3, eGFP-tagged full-length talin, $His_6SUMO$-kindlin2, as well as the generation of the Twin-Strep-tag vector for bacterial expression has been described in detail [50]. The generation of CEA3-ITGB3-CT fusion constructs has been described previously [28]. cDNA of human paxillin isoform a (NM_002859.4) was kindly provided by Alexander Bershadsky (Mechanobiology Institute, National University of Singapore, Singapore) and was used as template for polymerase chain reaction (PCR) amplification. $His_6$-SUMO-PXN LIM2/3 was generated by amplifying paxillin using primers: PXN LIM2/3 forward: 5′-CCAGTGGGTCTCAGGTGGTTCCCCGCGCTGCTAC-3′; PXN LIM2/3 reverse: 5′-CTGATCCTCGAGTTACCCATTCTTGAAATATTCAGGCGAGCCGCGCCGCTC-3′. The product was then ligated into pET24a His-Sumo bacterial expression vector using Eco31I and XhoI restriction sites.

Paxillin full length was amplified using primers PXN-fl forward: 5′-ACTCCTCCCCCGCCATGGACGACCTCGACGCCCTGCTG-3′ and PXN-fl reverse: 5′-CCCCACTAACCCGCTAGCAGAAGAGCTTGAGGAAGCAGTTCTGACAGTAAGG-3′. Paxillin LD1-5 was amplified using primers PXN-fl forward and PXN LD1-5 reverse: 5′-CCCCACTAACCCGCAGCTTGTTCAGGTCAG-3′. Paxillin LIM1-4 was amplified using primers PXN-LIM1-4 forward: 5′-ACTCCTCCCCCGCCATGAAGCTGGGGGGTCGCCACAGTCGCCAAAG-3′ and PXN-fl reverse.

cDNAs encoding LIM domain proteins: Hic-5 cDNA (isoform 1, NM_001042454.3) was kindly provided by Nicole Brimer (University of Virginia, Charlottesville, USA). Hic-5 was amplified using the primers Hic-5 forward: 5′-

ACTCCCCCGCCATGGAGGACCTGGATGCCC-3′ and Hic-5 reverse: 5′- CCCCACTAAC CCGTCAGCCGAAGAGCTTCAGG-3′. Leupaxin was amplified from pOTB7-LPXN (obtained from Harvard Medical School PlasmID Database; HsCD00331641) using primers LPXN forward: 5′- ACTCCTCCCCCGCCATGGAAGAGTTAGATGCC-3′ and LPXN reverse: 5′- CCCCACTAACCCGGCATTACAGTGGGAAGAGC-3′. PINCH-1 was amplified from pDNR-LIB hLIMS1 (obtained from Harvard Medical School PlasmID Database, HsCD00326503) using the primers PINCH forward: 5′-ACTCCTCCCCCGCCATGGCCAA CGCCCTGGCCAGC-3′ and PINCH reverse: 5′-CCCCACTAACCCGTTTCCTTCCTAAGG TCTCAGC-3′. cDNA for LASP-1 was provided by Elke Butt (Universitätsklinikum Würzburg, Würzburg, Germany) and amplified using primers LASP forward: 5′- ACTCCTCCCCCGCC ATGAACCCCAACTGCGCC-3′ and LASP reverse: 5′- CCCCACTAACCCGTCAGATGGC CTCCACGTAGTTGG-3′.

The respective PCR products were cloned into the pDNR-Dual-LIC vector according to the ligation independent cloning (LIC) strategy. The sequence verified constructs were then sub-cloned into the expression vector pEGFP-C1 harbouring a loxP recombination site via Cre-Lox recombination.

mEmerald-Migfilin was a gift from Michael Davidson (Addgene #54182) and was used unmodified. pEGFP-Zyxin was described elsewhere [51].

## Site directed mutagenesis

The amino acid residue of interest was changed using the overlap-extension PCR mutagenesis procedure. Desalted oligonucleotide primers were purchased from Sigma-Aldrich (Merck KGaA, Darmstadt, Germany) and can be found in following table. Human Paxillin LIM2/3 was used as a template.

Mutation Forward primer Reverse primer
Paxillin LIM2/3 F475A gcaGTGAACGGCAGCTTCTTC TGGCGTGAAGCATTCCC
Paxillin LIM2/3 V476A gcaAACGGCAGCTTCTTCGAGC GAATGGCGTGAAGCATTCC
Paxillin LIM2/3 N477A gcaGGCAGCTTCTTCGAGCAC CACGAATGGCGTGAAGCATTCC
Paxillin LIM2/3 G478A gcaAGCTTCTTCGAGCACGACG GTTCACGAATGGCGTGAAGC
Paxillin LIM2/3 S479A gcaTTCTTCGAGCACGACGG GCCGTTCACGAATGGCGTG
Paxillin LIM2/3 F480A gcaTTCGAGCACGACGGGCAG GCTGCCGTTCACGAATGG
Paxillin LIM2/3 F481A gcaGAGCACGACGGGCAGCCCTAC GAAGCTGCCGTTCACGAATGGC
Paxillin LIM2/3 4A tgcagctTTCTTCGAGCACGACGG gctgCGAATGGCGTGAAGCATTC

## Recombinant protein expression

Recombinant proteins with His$_6$-SUMO tag encoded on a pET24a vector were expressed in *E. coli* Tuner (DE3) cells. Cells were cultured in lysogeny broth medium containing 50 μg/mL kanamycin and 1% glucose (wt/vol) (for paxillin constructs additionally 0.1 mM ZnCl$_2$) at 37˚C until an OD$_{600}$ value of 0.6 to 0.8 was reached. Subsequently, overexpression was induced by addition of isopropyl β-D-thiogalactoside (IPTG) to a final concentration of 0.5 mM (for integrin β1 1 mM, respectively). After 6- to 8-h incubation (for integrin-β1 overnight, respectively) at 30˚C, cells were harvested by centrifugation at 10,000 × *g* for 15 min at 4˚C and stored at −80˚C. For isotopic labelling, bacteria were cultured in M9-minimal medium

containing $^{15}N$ ammonium chloride and/or $^{13}C$ glucose as sole nitrogen and carbon sources. Integrin cytoplasmic domains with a TwinStrepTagII tag encoded on a pET24a vector were expressed in *E. coli* BL21(DE3) pRosetta cells. Cells were cultured in lysogeny broth medium containing 50 μg/mL kanamycin. Expression conditions were identical to $His_6$-SUMO integrin β cytoplasmic tails. $His_6$-SUMO, $His_6$-SUMO-tagged talin F3, and kindlin2 were expressed in *E. coli* BL21(DE3). Bacteria were grown at 37˚C to an OD of 0.6 to 0.8 and induced with 1 mM IPTG overnight at 30˚C ($His_6$-SUMO and talin F3) or 20˚C (kindlin2).

## Protein purification

All steps were performed at 4˚C. Pelleted cells were slowly thawed on ice, resuspended in 1:5 (wt/vol) lysis buffer (50 mM Tris, 300 mM NaCl (pH 8.0), protease inhibitors), and lysed via a high-pressure homogenizer (Emulsiflex C3, Avestin, Ottawa, Canada). The mixture was separated by ultracentrifugation at $100,000 \times g$ for 30 min at 4˚C, and supernatant was loaded onto a HisTrap HP column (GE Healthcare, Freiburg, Germany) preequilibrated with 50 mM Tris, 10 mM imidazole and 300 mM NaCl (pH 8.0). The loaded column was washed and eluted fractions, monitored by UV absorbency at 280 nm, were pooled and dialyzed in 50 mM Tris, 300 mM NaCl (pH 8.0). After overnight cleavage with Ulp1, the $His_6$-SUMO tag was removed by subsequent HisTrap purification and protein solution was subjected to size-exclusion by using HiLoad 16/60 Superdex 30 (for integrin constructs) or Superdex 75 column (for paxillin constructs, GE Healthcare, Freiburg, Germany) preequilibrated with 50 mM $Na_2HPO_4$, 150 mM NaCl (pH 6.2) (for integrin constructs) or 7.5 (for paxillin constructs). For paxillin constructs, all buffers contained also 0.1 mM $ZnSO_4$ and 1 mM DTT, respectively. Purified protein was checked by SDS-PAGE.

## Pulldown assays with integrin β cytoplasmic domains

Approximately 2.5 μg of TwinStrep-tagged integrins or 10 μg of biotin-integrin peptides (β3wt aa742-788 (Biotin-HDRKEFAKFEEERARAKWDTANNPLYKEATSTFTNITYRGT-OH), β3Δ3aa aa742-785 (Biotin-HDRKEFAKFEEERARAKWDTANNPLYKEATSTFTNITY-OH), β3Δ8aa aa742-780 (Biotin-HDRKEFAKFEEERARAKWDTANNPLYKEATSTFT-OH); all from Novopep Limited) were loaded onto Strep-Tactin Sepharose beads (50% suspension; IBA Lifesciences) or streptavidin agarose beads (50% suspension; 16–126; Merck) in pulldown buffer (50 mM Tris (pH 8), 150 mM NaCl, 10% glycerol, 0.05% Tween, 10 μM $ZnCl_2$) for 30 min at RT under continuous rotation. After centrifugation (2,700$g$, 2 min, 4˚C), samples were washed 3 times with pulldown buffer. Then, integrin-loaded beads were suspended in bait protein solution (2 μM of protein diluted in pulldown buffer) and incubated 2 h at 4˚C under constant rotation. Samples were centrifuged (2,700$g$, 2 min, 4˚C) and washed 3 times with pulldown buffer. Strep-Tactin samples were eluted under native conditions by adding 30 μl of buffer BXT (50 mM Tris (pH 8), 150 mM NaCl, 50 mM biotin). After 10-min incubation at RT under constant rotation, samples were centrifuged. Supernatants were mixed with 4× SDS and boiled for 5 min at 95˚C before they were subjected to WB. Streptavidin agarose beads were directly mixed with 2× SDS and boiled for 10 min at 95˚C to elute proteins from biotin-integrin peptides before they were subjected to WB.

## Resonance assignment

All NMR-experiments for the resonance assignment and structure determination were recorded on a Bruker Avance III 600 MHz spectrometer equipped with an H/C/N TCI cryoprobe. Three-dimensional spectra were recorded using nonuniform sampling (25% to 50% sparse sampling) and reconstructed by recursive multidimensional decomposition (Topspin

v3.1–3.2). NMR-sample conditions: 500 μM $^{13}$C-$^{15}$N-Paxillin-LIM2/3, 150 mM NaCl, 50 mM Na$_2$HPO$_4$, 4 mM NaN$_3$, 1 mM DTT, 5% (or 100% D$_2$O) (pH 7.5). Recorded 3D-spectra in 5% D$_2$O: HNCO, HN(CA)CO, CBCANH, CBCA(CO)NH, H(CCCO)NH, (H)C(CCO)NH, NOE-SY-$^{15}$N-HSQC, NOESY-$^{13}$C$_{ali.}$-HSQC; in 100% D$_2$O: H(C)CH-TOCSY, (H)CCH-TOCSY, H(C)CH-COSY, NOESY-$^{13}$C$_{ali.}$-HSQC, NOESY-$^{13}$C$_{aro.}$-HSQC (NOESY mixing time: 120 ms in all spectra). Backbone resonance assignment was done semiautomatically using CARA v1.8.4.2 and Autolink II v0.8.7 [52]. The sidechain resonances were assigned manually. A nearly complete backbone assignment was achieved (99%) while N477 and N505 did not show amide resonances probably due to exchange broadening. The extent of the resonance assignment over all was 90% (see also Table 1). NOESY cross-peaks were picked and quantified using ATNOS [53] (implemented in UNIO'10 v2.0.2 [54]). TALOS-N was used to calculate φ- and ψ-angles based on the backbone chemical shifts [55]. The resonance assignment of paxillin LIM2/3 has been deposited to the BMRB (Entry 51154).

## Structure calculation

Initial structure calculation was done using Cyana v3.0 [56], with the protein sequence, the resonance assignment (CARA), NOESY-peaklists (ATNOS), and backbone angular restraints

**Table 1. Resonance assignment and structure determination statistics.**

| | |
|---|---|
| **Resonance assignment** | |
| *Backbone resonances* | $^1$H: 99%; $^{13}$C: 99%; $^{15}$N: 97% |
| All resonances | $^1$H: 91%; $^{13}$C: 92%; $^{15}$N: 77% |
| **NMR constraints** | |
| *Total unambiguous distance restraints* | 1,226 (100.0%) |
| Intraresidue (i, i) | 303 (24.7%) |
| Sequential (i, i+1) | 354 (28.9%) |
| Medium-range (2≤\|i-j\|≤4) | 151 (12.3%) |
| Long-range (\|i–j\|>4) | 418 (34.1%) |
| *Total dihedral angle restraints* | 343 |
| φ | 100 |
| ψ | 100 |
| χ$_1$ | 99 |
| χ$_2$ | 31 |
| χ$_3$ | 31 |
| **Ensemble statistics (20 structures)** | |
| *Violation analysis* | |
| Maximum distance violation (Å) | 0.61 |
| Maximum dihedral angle violation region (deg.) | 12.19 |
| *Target function* | |
| Mean CYANA target function | 5.75 ± 0.5 |
| *RMSD from mean structure* | |
| Backbone heavy atoms (Å) | 0.77 ± 0.17 (LIM2, P381-F438) 0.78 ± 0.30 (LIM3, P440-R497) |
| All heavy-atoms (Å) | 1.20 ± 0.16 |
| *Ramachandran plot* | |
| Most-favorable regions (%) | 77.4 |
| Additionally allowed regions (%) | 21.3 |
| Generously allowed regions (%) | 1.0 |
| Disallowed regions (%) | 0.4 |

(TALOS-N) as input. In later stages of the calculation, additional distance and angular constraints for a tetrahedral Zinc-coordination of the respective amino acids were implemented according to [57]. The coordination mode of the 4 histidines (H403, H406, H462, and H492) was determined by the difference of chemical shifts of $C^{\delta 2}$ and $C^{\varepsilon 1}$ [58], and in all cases, $\delta$ $(C^{\varepsilon 1}) - \delta$ $(C^{\delta 2})$ was larger than 17 ppm indicating a coordination via $N^{\delta 1}$ for all histidines. The structures were visualized and analyzed with PyMOL v1.3 (The PyMOL Molecular Graphics System, Version 2.0 Schrödinger, LLC). The coordinates of the final ensemble have been deposited to the PDB (Accession code: 7QB0)

## Chemical shift perturbation mapping

$^1$H-$^{15}$N HSQC spectra were recorded on a Bruker Avance III 600 MHz spectrometer equipped with a 5-mm BBI probe and a Bruker Avance NEO 500 MHz equipped with a H/C/N TCI CryoProbe Prodigy (Bruker Biospin GmbH, Rheinstetten, Germany). Chemical shifts were referenced to internal sodium 3-(Trimethylsilyl)propane-1-sulfonat-d$_6$ (DSS) at 0.0 ppm. The spectra were processed and analyzed with Topspin v2.1–4.0 (Bruker Biospin GmbH, Rheinstetten, Germany) and CARA (v. 1.9.1.5.). For NMR-experiments, the proteins were concentrated by repeated ultrafiltration (Amicon Ultra-4 Ultracel-3 kDa centrifugal filter device, Merck Millipore, Burlington, USA).

Experiments were performed at 298 K in buffer containing 50 mM Na$_2$HPO$_4$, 150 mM NaCl, 0.1 mM ZnSO$_4$, 1 mM DTT, 5% D$_2$O (pH = 6.2) (if integrin was $^{15}$N-labeled) or pH = 7.5 (if paxillin was $^{15}$N-labeled). Experimental procedure: To a sample of $^{15}$N-labeled protein, a stock solution of unlabeled interaction partner up to a final concentration of the respective constructs was added for collecting $^1$H-$^{15}$N HSQC spectra. The resonance assignment of integrin β1 was transferred from BMRB entry 16159 [59] and for integrin β3 from the BMRB entry 15552 [60]. Chemical shift change (Δδ) was calculated with the equation

$$\Delta\delta = \sqrt{0,5*[\Delta\delta_H^2 + 0,14*\Delta\delta_N^2]}$$

where Δδ [ppm] = δ$_{bound}$ − δ$_{free}$. The titration curves were fitted in OriginPro (v. b9.5.5.409) using the equation

$$\Delta\delta([L],[P]) = \Delta\delta_{max} \frac{([P]+[L]+K_D) - \sqrt{([P]+[L]+K_D)^2 - 4[P][L]}}{2[P]}$$

A simultaneous fit for multiple signals was used, allowing individual Δδ$_{max}$ values for each residue, but a global value for the dissociation constant K$_D$.

## sgRNA design and cloning

For the generation of recombinant sgRNA-expression vectors, we equipped the pBluescript vector (pBS SK+, Agilent Technologies, Santa Clara, CA, USA) with the murine U6 promotor controlled sgRNA expression cassette from pSpCas9(BB)-2A-GFP (PX458, a gift from Feng Zhang, Addgene plasmid # 48138) [61]. Therefore, we amplified the U6 controlled sgRNA expression cassette by PCR with the following primer pair: U6_sgRNA_forward: 5′-ATAGG TACCGTGAGGGCCTATTTCCC-3′ U6_sgRNA_reverse: 5′-ATACTCGAGGTCTGCAGA ATTGGCGC-3′. The resulting construct was cloned into pBS SK+ via XhoI and KpnI restriction sites. The sequence verified construct (pBS-U6) was then digested with BbsI and ligated with the annealed primer pair

MCS_oligo_forward:

5′-CACCGGGTCTTCGATGGGCCCAATTCGAATACACGTGGTTGATTTAAATGGG
CCCGAAGACCT-3′

MCS_oligo_reverse:

5′-AAACAGGTCTTCGGGCCCATTTAAATCAACCACGTGTATTCGAATTGG
GCCCATCGAAGACCC-3′

to create pBS-U6 with a multiple cloning site (pBS-U6-MCS) within the BbsI restriction sites. To generate the respective pBS-U6-Cer-sgRNA plasmid, the following sgRNA Oligos Cer-KO forward: 5′-CACCGCCGTCCAGCTCGACCAGGA-3′ and Cer-KO reverse: 5′-AAACTCCTGGTCGAGCTGGACGGC-3′ were annealed and ligated into the pBS-U6-MCS vector via the BbsI restriction sites. To eliminate remaining pBS-U6-MCS after the ligation step, samples were digested with BstBI. All constructs were sequence verified by LGC Genomics.

For targeting paxillin, the sgRNA oligos targeting exon 2 PXN-KO sense: 5′-CACCGAC GGTGGTGGTGGGACCGG-3′ and PXN-KO reverse: 5′-AAACCCGGTCCCACCACCA CCGTC-3′ were annealed and ligated into pSpCas9(BB)-2A-GFP (PX458-sgRNA mPXN)). For targeting murine integrin β3, the sgRNA oligos targeting exon 2 mITGB3-KO sense: 5′-C ACCGCGGACAGGATGCGAGCGCAG-3′ and mITGB3-KO reverse: 5′-AAACCTGCGCT CGCATCCTGTCCGC-3′ were annealed and ligated into pBS-U6-MCS to generate pBS-U6-mITGB3-sgRNA.

All sgRNAs were designed with the help of the CRISPR design tool (http://crispr.mit.edu) [62] and E-CRISP (www.e-crisp.org/E-CRISP).

## Generation of integrin β3 and paxillin knockout cell lines

For the generation of integrin β3 and paxillin KO cell lines, Flp-In 3T3 cell line (Invitrogen) was first stably transduced with a lentiviral vector encoding Histon2B mCerulean. Therefore, human histone 2B cDNA (H2B, gift from Thomas U. Meyer, University of Konstanz, Konstanz, Germany) was amplified by PCR with the following primer pair: hH2B_forward: 5′-ATAGCTAGCACCATGCCAGAGCCAGCGAAGTC-3′ hH2B_reverse: 5′-ATAACCGGTT TAGCGCTGGTGTACTTGG-3′ and cloned into pmCerulean-C1 (gift from David Piston, Vanderbilt University Medical Center, Nashville, USA) via NheI and AgeI restriction sites. The resulting construct was again subjected to PCR amplification with primers: H2B-Cer_forward: 5′-ATAGGATCCACCATGCCAGAGCCAGCGAAG-3′ and H2B-Cer_reverse: 5′-ATACTCGAGCTATTTGTACAGTTCGTCCATGCCG-3′. The PCR product was subcloned into pWZL Blasticidin (pWZLBlast, gift from Nicole Brimer, University of Virginia, Charlottesville, USA) via BamHI and XhoI restriction sites to generate pWZLBlast-H2B-Cer. For retroviral production, 80% confluent Phoenix-Eco cells [63] were transfected with pWZLBlast-H2B-Cer and cultured for 2 days. Afterwards, the supernatant was harvested, filtered through a 0.45-μm pore-size filter unit (Minisart, Sartorius Stedim Biotech GmbH, Göttingen, Germany) and applied on previously seeded NIH3T3 Flp-In cells at a ratio of 1:1 (vol/vol, supernatant: NIH3T3 growth medium) together with 4 μg/ml Polybrene (Sigma-Aldrich). Transduced cells were cultured in regular growth medium supplemented with 5 μg/ml blasticidin (Carl Roth GmbH + Co. KG, Karlsruhe, Germany). Cerulean-positive cells were sorted by FACS and seeded as single cells into 96-well plates to generate clonal Cerulean-positive NIH3T3 H2B-Cer Flp-In cell lines.

Paxillin KO cells were generated by transiently transfecting NIH3T3 H2B-Cer Flp-In cells with a combination of PX458-sgRNA mPXN + pBS-U6-Cer-sgRNA at a ratio of 1:5. Integrin β3 KO cells were generated by transiently transfecting NIH3T3 H2B-Cer Flp-In cells with a combination of PX458-sgRNA Cer + pBS-U6-mITGB3-sgRNA at a ratio of 1:5. Ten days after

transfection, single cerulean-negative cells were sorted into 96-well plates and clonal cell lines were expanded and KO of the target protein was verified by western blot.

## Stable complementation of knockout cells

For complementation of integrin β3, cDNA of murine integrin β3 (gift from Michael Davidson, Addgene plasmid # 54130) was amplified by PCR using the following primers mITGB3 forward: 5′-GATGACACTAGTGACCGCCATGCGAGCGCAGTG-3′ and mITGB3-fl reverse: 5′- TCGGCAGCCCTCGAGCTAAGTCCCCCGGTAGGTGATATTG-3′; mITGB3 forward and mITGB3Δ8aa reverse: 5′-TCGGCAGCCCTCGAGCTAGAAGGTGGAGGT GGCCTCTTTATAC-3′; mITGB3 forward and mITGB3Δ3aa reverse: 5′-TCGGCAGCCCTC GAGCTAGTAGGTGATATTGGTGAAGGTGGAGGTG-3′.

The respective products were cloned into pEF5/FRT-DEST (gift from Rajat Rohatgi, Addgene plasmid # 41008) using SpeI and PspXI restriction sites.

For complementation of Flp-In paxillin KO cells, we equipped the expression vector pEF5/ FRT-DEST with a GFP-tag adjacent to a LoxP site for C-terminal protein tagging via Cre-Lox recombination. Therefore, the respective sequence was amplified by PCR from pEGFP C1 (Clontech, Takara Bio Europe, Saint-Germain-en-Laye, France) with the following primer pair: EGFP_forward: 5′-GCCTAGACTAGTTAGCGCTACCGGTCGCCACCATG-3′ EGF-P_reverse: 5′-GCAGCGCTCGAGGGCTGATTATGATCAGTTATCTAGATCC-3′. The resulting construct was cloned into pEF5/FRT-DEST via SpeI and PspXI restriction sites to generate the expression vector pEF5/FRT EGFP C1 loxp.

The coding sequences (CDS) of paxillin was amplified by PCR with the following primers: PXN-fl forward: 5′-ACTCCTCCCCCGCCATGGACGACCTCGACGCCCTGCTG-3′ PXN-fl reverse: 5′- CCCCACTAACCCGCTAGCAGAAGAGCTTGAGGAAGCAGTTCTGACAGTAAG G-3′. PXN ΔLIM4 forward: 5′- ACTCCTCCCCCGCCATGGACGACCTCGACGC CCTGCTG -3′, PXN ΔLIM4 reverse: 5′-CCCCACTAACCCGCGAGCCGCGCCGCTC GTGGTAGTGC-3′; the paxillin 4A mutant was generated by overlap extension PCR. In a first PCR, 2 fragments were generated using primers PXN-fl forward and PXN-4A reverse: 5′- TGCTGCTGCAGCGAATGGCGTGAAGCATTCCCGGCACACAAAG -3′. For the second fragment primers PXN-4A forward: 5′- GCTGCAGCAGCATTCTTCGAGCACGACGGGC AGCCCTAC -3′ and PXN-fl reverse were used. In a second step, the 2 fragments were annealed by overlap extension PCR and amplified using primers PXN-fl forward and PXN-fl reverse.

The respective products were cloned into the pDNR-Dual-LIC vector according to the LIC strategy. The sequence verified constructs were then subcloned into the expression vector pEF5/FRT EGFP-C1 by Cre-Lox recombination.

Respective KO cell lines were complemented by transient transfection of 0.8 μg cDNA coding for the gene of interest + 3.2 μg Flp recombinase expression vector (pOG44) using jet-PRIME transfection reagent (Polyplus transfection, Illkirch, France). After 3 days, positive cells were selected by addition of 250 μg/ml Hygromycin B for 8 days.

## Flow cytometry

Cells were trypsinized and suspended in growth medium. Samples were centrifuged at 100$g$ for 3 min, and the resulting pellet was resuspended in FACS buffer (PBS with 5% FCS, 2 mM EDTA). Cells were washed once in FACS buffer, and $1 \times 10^6$ cells per sample were incubated with monoclonal anti-integrin antibodies as indicated for 1 h at 4°C under constant rotation. Cells were washed 3 times with FACS buffer, followed by incubation for 30 min with a

Rhodamine-Red–conjugated secondary antibody. Cells were analysed by flow cytometry (BD LSRFortessa, FACSDiva software, BD Biosciences, Heidelberg, Germany).

## Cell spreading analysis

Sterile glass coverslips were coated overnight at 4˚C with 5 μg/ml vitronectin or 5 μg/ml fibronectin type III repeats 9–11 (FNIII9-11). Cells were starved overnight in starvation medium (DMEM + 0.5% FCS). After 12-h starvation cells were trypsinized, trypsin was inactivated using soybean trypsin inhibitor (Applichem; 0.25 mg/ml in DMEM + 0.25% BSA). Cells were pelleted by centrifugation (100$g$, 3 min, RT) and suspended in DMEM + 0.25% BSA. Cells were kept in suspension for 30 min before seeding on coated glass coverslips. After 30 and 120 min of adherence, cells were washed once with PBS++ (0.5 mM $MgCl_2$, 0.9 mM $CaCl_2$), fixed with 4% PFA in PBS for 15 min at RT, washed thrice in PBS, permeabilized with 0.4% Triton-X-100 in PBS for 5 min at RT, washed thrice in PBS, and blocked for 30 min in blocking buffer (10% heat-inactivated CS in PBS). Cells were stained with CellMask Orange (diluted to 5 μg/ml in blocking buffer) and DAPI (diluted to 0.2 μg/ml in blocking buffer) for 30 min at RT. Images were analysed by a custom-built ImageJ macro. This macro has been deposited to the Zenodo database (https://zenodo.org/doi/10.5281/zenodo.12736436).

## Fluorescent microscopy and microscope settings

For confocal laser scanning microscopy, all images were taken from fixed specimens embedded in Dako fluorescent mounting medium (Dako, Carpinteria, USA) on a LEICA SP5 confocal microscope equipped with a 63.0×/1.40 NA oil HCX PL APO CS UV objective and analyzed using LAS AF Lite software. All images were acquired in xyz mode with 1,024 × 1,024 pixel format and 100 Hz scanning speed at 8-bit resolution. Fluorochromes used are Pacific Blue (excitation 405 nm, emission bandwidth: 435 to 475 nm); CF405M (excitation 405 nm, emission bandwidth: 435 to 475); GFP (excitation 488 nm, emission bandwidth: 500 to 525 nm); CellMask Orange (excitation 561 nm, emission bandwidth: 571 to 613 nm); RFP (excitation 561 nm, emission bandwidth 571 to 613 nm); and Cy5 (excitation 633 nm, emission bandwidth: 640 to 700 nm). Images were processed using ImageJ by applying the same brightness/contrast adjustments to all images within 1 experimental group.

## TIRF microscopy

Cells were starved overnight in starvation medium (DMEM + 0.5% FCS). After 12-h starvation cells were trypsinized, trypsin was inactivated using soybean trypsin inhibitor (Applichem; 0.25 mg/ml in DMEM + 0.25% BSA). Cells were pelleted by centrifugation (100$g$, 3 min, RT) and suspended in DMEM + 0.25% BSA. Cells were kept in suspension for 30 min before seeding on Wilco dishes, coated with 5 μg/ml vitronectin. Cells were imaged with a GE DeltaVision OMX Blazev4, equipped with a 60×/1.49 UIS2 APON TIRFM objective in Ring TIRF mode. Settings were adjusted to reach clean TIRF illumination without epifluorescence. A separate sCMOS camera was used for each channel, and images were later aligned using OMX image alignment calibration and softWoRx 7.0. Fluorophores used were GFP (excitation wavelength 488 nm, emission bandwidth: 528/48 nm) and Cy5 (excitation wavelength 647 nm, emission bandwidth: 683/40 nm).

## Opa-protein triggered integrin clustering (OPTIC)

OPTIC was performed as described previously [28]. Briefly, 293T cells were transfected with pcDNA3.1 CEACAM3-ITGB fusion constructs together with cDNA coding for the protein of

interest fused to eGFP. Cells were seeded on coverslips coated with 10 μg/ml poly-L-lysine in suspension medium (DMEM + 0.25% BSA), 48 h post-transfection. After 2 h, adherent cells were infected with Pacific Blue–stained *Neisseria gonorrhoeae* (Opa$_{52}$-expressing, non-piliated *N. gonorrhoeae* MS11-B2.1, kindly provided by T. Meyer, Berlin, Germany) at MOI 20 for 1 h in suspension medium. After 1 h, cells were fixed for 15 min with 4% paraformaldehyde in PBS at room temperature followed by 5-min permeabilization with 0.1% Triton X-100 in PBS. After washing with PBS, cells were incubated for 10 min in blocking solution (10% heat-inactivated calf serum in PBS) and stained for CEACAM3. After washing, cells were again incubated for 10 min in blocking solution followed by secondary antibody staining. Coverslips were mounted on glass slides using Dako fluorescent mounting medium (Dako, Carpinteria, USA).

## Supporting information

**S1 Fig. Paxillin and closely related LIM-domain proteins localize to clustered integrin-β1 or -β3 ct.** (**A**) Schematic overview of the OPTIC workflow. Opa-expressing Ngo are used to cluster CEACAM3-integrin β cytoplasmic tail fusion proteins potentially resulting in the recruitment of an intracellular protein of interest (POI). (**B**) 293T cells were transiently cotransfected with a CEACAM3-ITGB1 (ITGB1) or CEACAM3-ITGB3 (ITGB3) fusion construct together with GFP-labelled LIM-domain-containing proteins. WCLs of the transfected 293T cells were probed by western blotting with a monoclonal antibody against GFP to detect the expression of GFP-LIM proteins (upper panel). Coomassie staining (lower panel) was used to verify equal loading of the membrane. (**C**) 293T cells transfected with CEACAM3-ITGB1 (ITGB1) and the indicated GFP-fusion proteins were seeded on poly-L-lysine. Cells were infected for 1 h with Pacific Blue–labelled *Neisseria gonorrhoeae* (Ngo; blue), fixed, and stained for ITGB1 (red). Recruitment of GFP-LIM proteins to clustered ITGB1 tail is indicated by white arrowheads. Bars represent 2 μm. (**D**) Quantification of (C). Each data point reflects the recruitment ratio R in a CEACAM3-ITGB1-expressing cell with associated bacteria. Horizontal lines indicate mean values and 95% confidence intervals (whiskers) of $n = 60$ cells from 3 independent experiments. Statistical significance was calculated using one-way ANOVA, followed by Bonferroni multiple comparison test (*** $p < 0.001$, ns = not significant). The data underlying this panel can be found in S1 Data. (**E**) Quantification and statistical evaluation as in (D) of GFP-fusion protein recruitment to CEACAM3-ITGB3 (ITGB3) (see main Fig 1A). The data underlying this panel can be found in S1 Data.
(PDF)

**S2 Fig. Zinc fingers of the paxillin LIM3 domain show increased structural flexibility and are highly conserved across species.** (**A**) Heteronuclear $^{15}$N{$^{1}$H} NOE. The intensity ratio between spectra with and without $^{1}$H saturation is displayed vs. residue number. Values of 0.8 indicate rigid parts of the structure. Values smaller than 0.8 indicate increasing flexibility on the ps-to-ns timescale. The data underlying this panel can be found in S1 Data. On top of the heteronuclear NOE plot, the domain and secondary structure arrangement of the paxillin construct used in this study is depicted. α-Helices are shown in light blue. β-Sheets as magenta arrows. The secondary structure elements depicted here were identified with Pymol's dss command and correspond to the cartoon representation of the 3D structural ensemble shown in Fig 2. The linker connecting the LIM2 with the LIM3 domain is shown in green. Regions harboring flexible loops are shown in blue and grey, respectively. (**B**) Sequence alignment of paxillin LIM3 domain across different species: Alignment was performed using the structural alignment tool from T-Coffee and coloured using the BoxShade tool at ExPASy. Identical residues are shaded in black; highly similar residues are shaded gray. In addition, an alignment is shown of paxillin's LIM2 and LIM3 domains. The region harboring a flexible loop, which is

crucial for binding of the LIM3 domain to β-integrin, is indicated by a blue line.
(PDF)

**S3 Fig. The cytoplasmic domains of integrin-β1 and integrin-β3 support direct binding of paxillin.** (**A**) $^{15}$N-HSQC titration of 300 μM $^{15}$N integrin β1 ct (ITGB1 ct) with paxillin LIM2/3 (PXN LIM2/3). Paxillin was added in concentrations up to 500 μM. Boxes show a selection of signals affected by CSPs (residues K784, T788, and T789) in the presence of 0 μM (black), 150 μM (green), 300 μM (blue), and 650 μM (red) paxillin LIM2/3. Insets show the concentration dependence of combined amide CSPs globally fitted to a one site binding model. (**B**) Combined amide CSPs of 300 μM $^{15}$N integrin β1 ct in the presence of 650 μM paxillin LIM2/3 vs. residue number of integrin β1 ct. (**C**) $^{15}$N-HSQC titration of 300 μM $^{15}$N paxillin LIM2/3 (PXN LIM2/3) with integrin β3 ct Δ8aa (ITGB3 Δ8aa). Integrin was added up to a concentration of 600 μM. (**D**) Combined amide CSPs of 300 μM $^{15}$N paxillin LIM2/3 in the presence of 600 μM integrin β3 ct Δ8aa vs. residue number of paxillin LIM2/3. (**E**) 293T cells were transiently cotransfected with a CEACAM3 ITGB3 (CEA3-ITGB3) fusion construct or the indicated truncated ITGB3 mutants together with GFP or GFP-paxillin and seeded on poly-L-lysine. Cells were infected for 1 h with Pacific Blue–labelled *Neisseria gonorrhoeae* (Ngo, blue), fixed, and stained for ITGB3 (red). Recruitment of GFP-Paxillin to clustered ITGB3 tails is indicated by white arrowheads. Bars represent 1 μm. (**F**) Quantification of GFP/GFP-paxillin recruitment to the indicated CEA3-ITGB3 variants from (E). Shown are means and 95% confidence intervals of $n$ = 60 cells from 3 independent experiments. Significance was calculated using one-way ANOVA followed by Bonferroni multiple comparison test. Significance levels compared to paxillin wt are indicated (ns: not significant; *** $p \leq 0.001$). The data underlying this panel can be found in S1 Data. (**G**) Streptactin pulldown of recombinant His-SUMO or His-SUMO-talin F3 using the Strep-tag integrin β3 cytoplasmic tail in the wt form or with a truncation of the carboxy-terminal 3 (Δ3) or 8 (Δ8) amino acids. Integrin β3–associated talin F3 domain was detected by anti-His-tag blot (upper panel); the Strep-tag integrin tails were revealed by streptactin blot (lower panel). The input of purified His-SUMO or His-SUMO-Talin F3 protein is shown on the left hand side. (**H**) NMR-based interaction study between talin's FERM domain and integrin β3. Superposition of $^{15}$N-HSQC spectra of $^{15}$N-labeled wt ITGB3 (left graph) or ITGB3 Δ3 (ΔRGT; right graph). In both experiments, multiple signals of the integrin peptide shift or disappear completely through the interaction with the large FERM domain, indicating equivalent binding of the wt and Δ3 integrin peptide to talin.
(PDF)

**S4 Fig. NMR experiments with paxillin LIM3 mutants show that conserved aromatic residues in the flexible loop are essential for maintaining a stably folded structure.** Alanine scan of paxillin LIM3's flexible loop region. Shown are superpositions of $^1$H-$^{15}$N-HSQC spectra of wt paxillin LIM2/3 (black) and paxillin LIM2/3 mutants (red). The bar graphs below show the combined amide CSPs of $^{15}$N-labelled paxillin LIM2/3 wt compared to the indicated mutant paxillin LIM2/3 along the amino acid sequence. (**A**) Paxillin LIM2/3 F475A. (**B**) Paxillin LIM2/3 F480A. (**C**) Paxillin LIM2/3 F481A. (**D**) Paxillin LIM2/3 V476A. (**E**) Paxillin LIM2/3 S479A. (**F**) Paxillin LIM2/3 4A.
(PDF)

**S5 Fig. The direct interaction between the paxillin LIM3 flexible loop and integrin β3 contributes to cell spreading.** (**A**) NIH 3T3 Flp-In cells (NIH3T3) were used to derive paxillin KO cells. Paxillin KO cells were stably transfected with the empty vector (PXN KO) or vectors encoding GFP-paxillin wt (PXN wt), paxillin lacking the LIM4 domain (PXN ΔLIM4), or paxillin with a mutated flexible loop in LIM3 (PXN-4A). Western blot of WCLs with anti-paxillin

antibody demonstrates the lack of endogenous paxillin in the PXN KO cells and reexpression of equivalent levels of paxillin wt or mutants in the stably reexpressing cells (upper panel). The lower panel verifies equal loading of samples by anti-tubulin blot. (**B**) Cells from (A) were starved overnight and seeded for 30 or 120 min, respectively, on the integrin ligand fibronectin in the absence of serum. Cells were fixed and the cell membrane was stained with CellMask Orange. Scale bar represents 20 μm (upper panel). To analyse cell spreading, the area of individual cells was quantified (lower panel). Shown are mean values with 95% confidence intervals from 3 independent experiments for 30-min time point or from 2 independent experiments for 120-min time point. Sample sizes are given in brackets. Statistical significance was calculated using one-way ANOVA followed by Bonferroni multiple comparison test (ns: not significant; *** $p \leq 0.001$; ** $p \leq 0.01$). The data underlying this panel can be found in S1 Data. (**C**) TIRF microscopy images of PXN KO and reexpressing cell lines seeded on vitronectin for 30 min before fixation. Cells were stained with a monoclonal anti-kindlin antibody. (**D**, **E**) FA analysis of $n > 1,000$ kindlin-positive FAs per sample. FAs were analysed for GFP-paxillin intensity (**D**) and FA area (**E**). Violin plot in (E) show the Kernel probability density distribution. Densities are plotted symmetrically to the left and right of the box plots. Inserted boxplots show mean values with 95% confidence intervals. Statistical significance was calculated using one-way ANOVA followed by Bonferroni multiple comparison test. Significance levels compared to paxillin wt are indicated (ns: not significant; *** $p \leq 0.001$; ** $p \leq 0.01$; * $p \leq 0.05$). Boxplots in (D) show mean and error bars represent 5 and 95 percentiles. Significance was calculated using one-way ANOVA, followed by Bonferroni multiple comparison test (*** $p < 0.001$, ns = not significant). (**F**) NIH Flp-In integrin β3 KO and the indicated ITGB3 wt, ITGB3 Δ8aa, or ITGB3 Δ3aa reexpressing cell lines were serum starved and seeded onto fibronectin-coated (5 μg/ml) glass slides for 30 min and cell area was measured. Shown are mean values and 95% confidence intervals of $n = 60$ cells per sample from 3 independent experiments. Statistical significance was calculated using one-way ANOVA followed by Bonferroni multiple comparison test (*** $p \leq 0.001$; ** $p \leq 0.01$). The data underlying panels D–F can be found in S1 Data.
(PDF)

**S1 Data. All numerical values underlying this study.**
(XLSX)

**S1 Raw Images. Raw images of Figs 1F, 2F, 3E, 4B, 4E, S1B, S3G and S5A.**
(PDF)

## Acknowledgments

We gratefully acknowledge initial contributions to peptide and protein preparation and interaction studies by J. Ude, M. Roth, M. Gallandi, and S. Feindler-Boeckh, as well as expert support in mass spectrometry by Dr I. Starke. We thank D. Schlaepfer (UCSD, San Diego, CA) for providing GFP-FAK reexpressing FAK−/− murine fibroblasts and R. Fässler and R. Böttcher (MPI for Biochemistry, Martinsried, Germany) for providing kindlin1/kindlin2-deficient mouse fibroblasts. The authors thank Alexander Bershadsky (Mechanobiology Institute, National University of Singapore, Singapore), Nicole Brimer (University of Virginia, Charlottesville, USA), and Elke Butt (Universitätsklinikum Würzburg, Würzburg, Germany) for providing constructs. We would also like to thank the Core Facilities of the University of Konstanz for excellent help and support with cell sorting (A. Sommershof, FlowKon) and microscopy (M. Stöckl, Bioimaging Center).

## Author Contributions

**Conceptualization:** Timo Baade, Marcus Michaelis, Christof R. Hauck, Heiko M. Möller.

**Funding acquisition:** Christof R. Hauck, Heiko M. Möller.

**Investigation:** Timo Baade, Marcus Michaelis, Andreas Prestel, Christoph Paone, Nikolai Klishin, Marleen Herbinger, Laura Scheinost.

**Methodology:** Timo Baade, Marcus Michaelis, Andreas Prestel, Christoph Paone, Nikolai Klishin, Marleen Herbinger, Ruslan Nedielkov, Christof R. Hauck, Heiko M. Möller.

**Project administration:** Christof R. Hauck, Heiko M. Möller.

**Supervision:** Christof R. Hauck, Heiko M. Möller.

**Writing – original draft:** Timo Baade, Marcus Michaelis, Christof R. Hauck, Heiko M. Möller.

**Writing – review & editing:** Christof R. Hauck, Heiko M. Möller.

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
