## [Editor Report · Decision Letter 0]

11 Sep 2023

Dear Dr Hauck, 

Thank you for submitting your manuscript entitled "The solution structure of the paxillin LIM3 domain reveals a flexible loop mediating direct binding to integrin β3" for consideration as a Research Article by PLOS Biology.

Your manuscript has now been evaluated by the PLOS Biology editorial staff as well as by an academic editor with relevant expertise and I am writing to let you know that we would like to send your submission out for external peer review. However, we would like to consider the manuscript as a Short Report. Thus, please select that type of article from the dropdown menu when you submit the metadata (see below).

Before we can send your manuscript to reviewers, we need you to complete your submission by providing the metadata that is required for full assessment. To this end, please login to Editorial Manager where you will find the paper in the 'Submissions Needing Revisions' folder on your homepage. Please click 'Revise Submission' from the Action Links and complete all additional questions in the submission questionnaire.

Once your full submission is complete, your paper will undergo a series of checks in preparation for peer review. After your manuscript has passed the checks it will be sent out for review. To provide the metadata for your submission, please Login to Editorial Manager (https://www.editorialmanager.com/pbiology) within two working days, i.e. by Sep 13 2023 11:59PM.

Kind regards,

Richard

Richard Hodge, PhD

Senior Editor

PLOS Biology

rhodge@plos.org

---

## [Decision Letter · Decision Letter 1]

22 Nov 2023

Dear Dr Hauck,

Thank you for your continued patience while your manuscript "The solution structure of the paxillin LIM3 domain reveals a flexible loop mediating direct binding to integrin β3" was peer-reviewed at PLOS Biology. Please accept my sincere apologies for the long delays that you have experienced during the peer review process. Your manuscript has now been evaluated by the PLOS Biology editors, an Academic Editor with relevant expertise, and by three independent reviewers. 

In light of the reviews, which you will find at the end of this email, we would like to invite you to revise the work to thoroughly address the reviewers' reports.

As you can see, the reviewers generally think the findings are interesting for the field and note that the NMR analyses are well done. However, Reviewer #1 raises concerns the overall physiological relevance of the interaction and the proposed mechanistic model given the overlapping binding site of paxillin on integrin with kindlin. In the revised manuscript, we ask that you please discuss the problems and complexities in studying the repetitive LIM domains and the paxillin/kindlin vs paxillin/integrin interactions, as well as providing a more balanced and nuanced introduction along the lines proposed by Reviewer #3. After discussions with Academic Editor, we strongly encourage you to address the concerns of Reviewer #1 with additional experiments to enhance the physiological relevance of the study, but we will not make this essential to consider the revised version given the Short Report format.

Given the extent of revision needed, we cannot make a decision about publication until we have seen the revised manuscript and your response to the reviewers' comments. Your revised manuscript is likely to be sent for further evaluation by all or a subset of the reviewers.

**IMPORTANT - SUBMITTING YOUR REVISION**

*Re-submission Checklist*

*Published Peer Review*

*PLOS Data Policy*

*Blot and Gel Data Policy*

Sincerely,

Richard

Richard Hodge, PhD

rhodge@plos.org

REVIEWS:

Reviewer #1: This manuscript reports the NMR-based structural and functional studies of paxillin LIM2-3 domain and its interaction with integrin beta cytoplasmic tails. The major conclusion is that such interaction is crucial for recruiting paxillin to the integrin adhesion site independent of kindlin. While the interaction appears to be novel, the physiological relevance of this interaction and the authors' major conclusion are not supported by the data. It has been well established that talin and kindlin, both of which bind to paxillin as the authors indicated, directly bind to integrin and trigger focal adhesion assembly. In fact, it was shown clearly that kindlin, which binds paxillin at nM affinity (Böttcher RT et al., J Cell Biol, 2017) and recruits paxillin to focal adhesion (Theodosiou et al., eLife, 2016). Of course, talin-paxillin interaction also promotes paxillin recruitment to integrin site. These earlier data show clearly that paxillin coordinates with kindlin and talin to be recruited to the integrin site not independently contrasting to what the authors proposed here. The technical flaws I saw in the authors' experiment are:

1. The paxillin LIM3 binding site on integrin tail overlaps with that of kindlin and probably part of talin too (Fig 2D) based on the talin-integrin and kindlin-integrin complex structures. Thus, the 3aa and 8aa deletion in integrin tail would clearly affect kindlin and likely talin too. These deletion data cannot support the paxillin recruitment to integrin site independent of kindlin and likely talin.

2. The mutations on LIM3 loop caused some cellular defects. However, given that LIM3 may also partially bind to other proteins such as Kindlin (Böttcher RT et al., J Cell Biol, 2017) and PTP-PEST (Cote et al., J. Biol. Chem, 274:20550-60, 1999), it is unclear if the mutations would also impair the interaction with these proteins, making the analysis ambiguous.

3. Kindlin, talin, and paxillin are all found in nascent focal adhesions. Given the overlapping binding site of paxillin on integrin with kindlin, it is hard to imagine how paxillin would localize to integrin independent of kindlin. The only possibility to me is that paxillin and kindlin bind to different integrins on the cell surface while being localized to integrin site but there is not data in the manuscript to support this possibility. 

Overall, while I think the in vitro binding analysis appears to be ok, the physiological relevance and the mechanism of such binding remain confusing.

Reviewer #2: The paper by Baade et al. Report on the study of interactions of the paxillin LlM3 domain with an integrin protein. The paper contains cell-biological and biophysical experiments to probe for the interaction. The paper is well written, interesting to read and the conclusions are backed by data. The necessary controls have been mostly made. I consider this a careful study that should be published, and I have not detected any major issues.

I am not an expert on the cell-biological experiments but rather want to comment on the NMR part. I have a few minor suggestions:

* To better comprehend please color-code the linker in Fig. 2

* More importantly, please show domain borders in the heteronuclear NOE data in Fig S2A. I assume that the linker is around residue 452 (?). However, I noticed 3 more regions of slightly increased flexibility, which is likely due to loops (around residues 392, 418 and 480). Only the latter is discussed in the text.

* Is the flexible loop in the LlM3 domain rigidified upon interaction with the integrin? Any change in the magnitude of the heteronuclear NOE for loop residues?

* How many interdomain contacts are observed as NOEs?

* What is the sequence homology of LlM2 and LlM3? Do they stem from gene duplications?

* Concerning the interaction studies: I realized that CSPs were much larger on the LlM domains than on ITGB3 (compare figures 2C and 3A). Please comment. Or does the interface involve aromatic residues on one side that would result in larger CSPs?

* Sometimes figures are differently in size. Fr example 3B and 3D show the same region on both axis but the figures are differently scaled so that a wrong impression about the relative CSPs is given, please make the plots identical in size.

* I always prefer to have an HSQC spectrum in the main paper because it tells a lot how well-behaved the protein is (fonts seem to be stretched in the horizontal direction in S2C)

* Which residues were used for the superposition for the calc of the bb RMSD in Table 1?

* Did the structure calculation comprise a refinement in explicit water? Which parameter were used as distances for the Zn coordination?

* Line 601: The authors name is "Güntert"

* Please insert a reference for the method to establish the coordination mode of the His residues

* Please add assignment statistics to Table 1 or elsewhere. Were all backbone atoms assigned?

* I am not an expert on this, but to which extent is a interaction of 530 uM biological significant (lines 203/204). Please comment.

* The equation on line 664 has a typo. The "2" after the round brackets under the root is a square, not a factor of 2.

Reviewer #3: Paxillin is one of the most important but also still highly enigmatic proteins, localizing to cell-matrix adhesions (also called focal adhesions), thereby controlling the critical mechanisms of cell spreading. Paxillin is composed to of a highly flexible N-terminal domain exhibiting several small structural-defined helical motifs with characteristic acidic and hydrophobic amino acids (LD-motifs). These LD-motifs can interact with other focal adhesion proteins such as talin, vinculin and focal adhesion kinase. Interestingly, these LD-motif interactions are considered of low-affinity, since the expression of the N-terminal domain alone, is not enabling the efficient recruitment of paxillin to integrin-containing focal adhesions. In addition the N-terminal domain contains a proline stretch forming a binding site for the src SH3 domain, potentially relevant for regulation of paxillin recruitment to focal adhesions by tyr-phosphorylation. In contrast to the highly flexible N-terminal domain, the focal adhesion-targeting domain of paxillin is localized in the c-terminal domain of paxillin, which is composed of 4 tightly spaced LIM-domains. Initial LIM-domain deletion studies by Brown and Turner (1996) have shown that deletions of LIM3 and to a minor level LIM2, cause a strong reduction in focal adhesion localization when transfected in fibroblasts. However, more recently, quantitative approaches to measure the individual contribution of the LIM-domains to beta3-integrin-containing focal adhesions, revealed a surprisingly non-sequence specific role of LIM3 in the paxillin recruitment mechanisms. The exchange of LIM3 with either LIM1 or LIM2, barely modified paxillin affinity to focal adhesions, while its deletion strongly affected it (Ripamonti et al., 2021). Instead of an adhesion targeting role in LIM3, a lipid-binding motif was detected in LIM4 and the strongest interference with focal adhesion recruitment was observed by exchanging the LIM2 and LIM1, with LIM3 domains. 

This very old and the more recent studies set the stage for the current manuscript, but highlights also potential problems in the interpretation of the biological results, when focusing exclusively on the role of the LIM3 domain for paxillin recruitment to focal adhesions. Due to the central position of the LIM3-domain in the 4 LIM-domain array, slight perturbation of the LIM3 domain will cause paxillin recruitment defect that cannot be easily distinguished from a perturbation with a flexible integrin peptide. While direct integrin/paxillin LIM-domain interactions were never shown before, the interactions of NPXY-motif containing peptides with the LIM-domains from Enigma, and from 4-1/2LIM-domain proteins were reported. Based on these previous observations, we would expect the NPXY-motifs of integrins to bind to paxillin LIM domains, potentially binding to hydrophobic pockets present on the LIM-domains.

Here in this manuscript the authors use NMR to probe the interaction of a purified LIM2/LIM3 tandem construct with either the beta1 and beta3 integrin tails. As expected the interactions are of low affinity, but sufficiently robust to evaluate the direct interaction of integrins with the LIM-domain of paxillin (surprisingly beta1 binds 10x better than beta3). By mutagenesis the authors identify a conserved and slightly flexible loop in LIM3 that is apparently relevant for binding to the very c-terminal end of the integrin b3-tail. They tested c-terminal deletions of 3 and 8 residues, in pull-down and cell-based assays with kindlin-deficient cells, as well as Ala-mutation of the loop residues in LIM3. When probing with N15-labeled integrin tails, perturbations by the Paxilin LIM2/LIM3 construct were found in the first and second NPXY-motif and at conserved inter-NPXY-residues for the beta3-tail, while for beta1 integrin tail, perturbations were identified only in the first NPXY-motif and inter-NPXY-residues, but not in the second NPXY-motif. Further c-terminal deletion of three residues in beta3, reduced paxillin pull-down, and completely blocked it when the 8 c-terminal residues were removed. Similarly paxillin 4A-substitution in the LIM3-loop only partially reduced beta3 binding, showing an alternative beta3 binding site in LIM2. Expression of beta3 integrins in kindlin ko cells partially rescued cell spreading, by developing small and instable cell projections. These projections were lost when the 3 and 8 c-terminal residues were removed, showing a more severe cellular phenotype compared to the pull-down and NMR-perturbation experiment, showing partial interactions with the del3-construct. Beta3 integrin-deletion studies indicated a signaling function for the last three amino acids in beta-3 integrins, which is consistent with the previously identified c-src binding site (however not mentioned in the manuscript). 

Based on these data the authors propose a conserved binding site for beta3-integrins in the LIM3 domain of paxillin, recognizing the C-terminal end of the integrin peptide. However, the direct link between paxillin and beta3 integrin binding at LIM3 is not that evident and the effect of c-terminal integrin deletions are not consistent with the effects of the pull-down assays. Instead, the data supports a critical signaling function of the c-terminal RGT sequence in beta-3, while showing only minor perturbation of paxillin pull-down. Similarly mutation of LIM3 leads to NMR perturbation in LIM2, while maintaining the integrin pull-down. Therefore the presented data are consistent with deletion of the src binding site in the c-terminal integrin tail, and a redundant or multiple LIM-domain spanning binding sites for integrin tails. In summary, the quality of presented data is very good, but in multiple positions, the drawn conclusions are not supported by the data. This requires a considerable rephrasing of the text, a better and more balanced introduction to the field and a more careful interpretation of the data.

Major points:

Introduction: 

While the general introduction into focal adhesions and the critical role of talin and kindlin is well done, it is important to mention that information gathered from studies in talin or kindlin-ko-cells are limited by their non-physiological context. Especially when Mn-dependent cell-matrix adhesions are introduced, the readers should be aware of the experimental nature of these adhesions, failing to represent the regulatory complexity of focal adhesions. For example, when citing Atherton et al., 2020, where the known link between the paxillin N-terminus and vinculin tail, or talin-rod domains has been experimentally confirmed, the readers should be made aware that in talin ko cells, paxillin and vinculin recruitment to Mn-activated integrins was independent from each other, which is different from what is stated in the text (line 119). In lines 123 -125, the idea that kindlin directs paxillin to focal adhesion is well introduced, but the alternative vision that the proximal NPXY-motif in integrins is critical for paxillin recruitment and induction of cell spreading is not introduced (Pinon et al., cited in the discussion). In addition, the recent literature analyzing the differential roles for LIM2 and LIM3 for recruitment of paxillin to focal adhesions should also be introduce (Ripamonti et al 2021; cited in the discussion). In this work, a critical and sequence specific role for LIM2, but not for LIM3 has been demonstrated. This information is particularly relevant, as the authors show partial reductions in integrin pull-downs with the 4A-mutant of the LIM2/LIM3 tandem construct, which proposes additional LIM-domain-dependent interactions outside of the LIM3 domain (e.g. Fig 3C-E). 

Results:

Fig1. This new bacteria-based integrin clustering experiments nicely shows the integrin specificity of the paxillin family of LIM-domain containing proteins. This is an important experiment for the community as integrin-dependent (this report) versus tension-dependent (Schiller et al) LIM-domain recruitment mechanisms can be distinguished.

Fig2. NMR-based titration experiments are a very powerful technique to analyze low-affinity interactions between peptides and rigid protein domains. A general problem with low-affinity interactions is however the very high concentration of protein required for the measurement. High protein concentrations can reveal non-physiological effects within the protein solution. I found it surprising that the NMR-perturbation signal in the different titration curves for beta-3 integrin tail in Fig2C (inserts), is not saturating. This is different for the analysis with the beta1-integrin tail, potentially reflected by the 10 fold different affinity constant. The addition of a sentence mentioning this non-saturating behavior would be important, as it points to multiple less-definable interaction sites. 

Fig2. The readers should be made aware of the discrepancy of the data shown in figure 2. While the NMR-perturbation experiments show a complete loss of interaction in the absence of the RGT-sequence in the beta3-integrin, there is still a considerable amount of LIM2/LIM3 pulled-down with the RGT-deleted integrin beta-3 peptide. Apparently, beta-3 integrin-peptide interactions at the first and second NPXY, as well as inter-NPXY-region are critically dependent on the RGT-peptide. Since Arg-residues are often involved in protein-protein or protein-peptide interactions, the specific mutation of the Arg-residue would have been a logic addition, especially since in beta1, a Glu-residue is following the second NPXY-motif, without inducing a relevant integrin/paxillin interaction. 

Fig.3. The NMR perturbation experiments with the N15-labeled LIM2/LIM3 tandem domain are very instructive. However, there is a circular argument hidden in the analysis, that should be better explained to the reader. While showing that point mutations in critical Phe-residues lead to entire LIM3-domain perturbation (see figure S4A and S4B; for F475A and F480A-mutant), the same F480-residue is flexible and its side chain can be found unusually solvent exposed when interacting with the integrin beta-3 peptide (Fig. 3B). Could this actually mean that the beta-3 peptide is denaturing the LIM3 domain in this loop, leading to extensive remodeling of the LIM3 domain structure. Unfortunately, the subsequently used 4A-mutation, is also affecting the NMR-signal of neighboring LIM3-residues quite broadly, very similar to the F480A mutation (Fig. S4F). Therefore, I am particularly concerned with the mutation of the Gly-residues in the 4A-mutant. Since folding of LIM-domains does not only require hydrophobic residues, but also requires tight turns formed by critically positioned gly-residues, it is well possible that the 4A-mutation is locally perturbing the folding of the LIM3 domain. Blocking the gly-residue with the 4A-mutation could have strong negative effects on the folding of the neighboring residues and affect the entre LIM3 structure in complicated ways. 

Moreover, figures 3C,D show that in the presence of the 4A-mutation in LIM3, there are new LIM2/beta3 interactions appearing, suggesting that the specificity of the detected beta3 interaction with LIM3 are rather low. This should be clearly stated in the manuscript, to avoid confusion and misinterpretation by the readers. 

The considerable LIM2 involvement in beta3-binding is also shown in Fig. 3E, where the pull down with the 4A-mutant shows still important interactions not consistent with a unique critical role of LIM3. 

Fig. 4. Two different knockouts strategies are used in this figure to show the importance of the integrin c-terminal domain and that of kindlin, known to bind to this domain. Unfortunately, this figure is disconnected from the rest of the paper and an obvious link to the previously analyzed paxillin/integrin interaction cannot be drawn. Deletion of the RGT-sequence in beta3-integrin reduces spreading, while completely blocking it in the 8 aa-deletion. Although not specifically analyzed, the deletion of RGT, may affect kindlin-recruitment, and kindlin-mediated cell spreading. Therefore causing a spreading defect because of reduced kindlin-binding rather than paxillin recruitment. This dependence on kindlin for cell spreading is confirmed by the removal of kindlin, blocking spreading of both the RGT- and 8aa- integrin deletion constructs. Thus kindlin is more relevant than a potential paxillin binding at the c-terminal integrin sequence.

Previous work by Theodosiou et al., 2016, has shown that in the absence of kindlin2, paxillin is no longer recruited to Mn-induced talin-containing adhesions (please also correct the section in the introduction, where it is incorrectly stated that paxillin is recruited to talin, in kindlin-ko cells, the reverse has been observed by Theodosiou). According to the pull-down in the previous figures, paxillin should still be recruited to kindlin-deficient beta-3 wt or delta-RGT-integrin expressing cells, which is not the case from the images shown in Fig. 4F (while talin accumulation can be seen in the projections in figure 4F (right panel). Thus the biochemical data is not directly translatable to the observations made in kindlin-ko cells, and this should be clearly said in the manuscript, in order to avoid overinterpretation of the data.

Minor points

In line 219, the beta3-S778A mutation is mentioned, without citing or showing the data that this mutation would induce a kindlin-binding defect. Likewise, potential kindlin-binding defects to the del RGT-sequence are not shown, but are key for understanding the experiments in figure 4. 

In line 244, the use of the paxillin delta-LIM4 construct is mentioned as control for a perturbed LIM4/kindlin interaction. Considering that the LIM4 deletion is mainly affecting membrane-interaction and has only modest effects on the recruitment of paxillin to focal adhesions, this control is not able to distinguish between a kindlin, or integrin-dependent paxillin recruitment mechanisms. 

When analyzing the flexibility of the LIM2/LIM3 construct by NMR, there is a similar flexible loop observable in LIM2 (Fig. S2). The authors should comment on this similarity, and discuss how solvent exposure of the flanking Phe-side chains as seen in the NMR structure, could perturb the overall folding of the LIM domains. 

In summary, the manuscript shows a series of very interesting observations that are helpful for analyzing the mechanisms of LIM-domain-dependent recruitment of paxillin to integrin-cytoplasmic tails. Unfortunately, this reviewer is under the impression that the fixation on the likely overstated role of LIM3 for paxillin recruitment has prevented the authors from reporting potentially relevant interactions with the LIM2-domain of paxillin. In addition to the kindlin-mediated recruitment of paxillin, the direct interaction with integrin peptides is still quite obscure (see also Pinon et al., 2014) and the specificity and role of a direct integrin interaction with paxillin needs to be analyzed together with a proper discussion of the literature analyzing integrin/paxillin dependent cell spreading. In this context, the observed differences between beta1 and beta3 integrins are very interesting and need to be further discussed, and linked to the reported association of src with the c-terminal beta3-integrin tail.

---

## [Decision Letter · Decision Letter 2]

20 Jun 2024

Dear Dr Hauck,

Thank you for your patience while we considered your revised manuscript "The solution structure of the paxillin LIM3 domain reveals a flexible loop mediating direct binding to integrin β3" for publication as a Short Reports at PLOS Biology. This revised version of your manuscript has been evaluated by the PLOS Biology editors, the Academic Editor and the original reviewers. Please accept my apologies for the unusual delay incurred in assessing your revision.

Based on the reviews, we are likely to accept this manuscript for publication, provided you satisfactorily address the remaining points raised by the reviewers. Please also make sure to address the following data and other policy-related requests.

IMPORTANT - please attend to the following:

*We would like to suggest a different title to improve accessibility. Please change your title to: "A flexible loop in the paxillin LIM3 domain mediates its direct binding to integrin β3" 

* Please note that per journal policy, the model system/species studied should be clearly stated in the abstract of your manuscript. Therefore, please mention the cell lines that you used in the abstract.

* Please take the funding disclosure out of the acknowledgments. These will be included in the paper based on your entry in Editorial Manager.

* DATA POLICY:

Regardless of the method selected, please ensure that you provide the individual numerical values that underlie the summary data displayed in the following figure panels as they are essential for readers to assess your analysis and to reproduce it: Figs 1CE, 2F, 3F, 4C, S1DE, S2A, S3F and S5BDEF.

* CODE POLICY

Please note that we cannot accept sole deposition of code in GitHub, as this could be changed after publication. However, you can archive this version of your publicly available GitHub code to Zenodo. Once you do this, it will generate a DOI number, which you will need to provide in the Data Accessibility Statement (you are welcome to also provide the GitHub access information). See the process for doing this here: https://docs.github.com/en/repositories/archiving-a-github-repository/referencing-and-citing-content.

We require the original, uncropped and minimally adjusted images supporting all blot and gel results reported in an article's figures or Supporting Information files. We will require these files before a manuscript can be accepted so please prepare and upload them now. Please carefully read our guidelines for how to prepare and upload this data: https://journals.plos.org/plosbiology/s/figures#loc-blot-and-gel-reporting-requirements

Please provide the original gels for: 1F, 2F, 3E, 4BE, S1B, S3G, S5A, S6BE.

We expect to receive your revised manuscript within two weeks. 

*Published Peer Review History*

*Press*

Sincerely,

Suzanne

Suzanne De Bruijn, PhD, 

Associate Editor

sbruijn@plos.org

PLOS Biology

Reviewer remarks:

Reviewer #2: all points I have raised in the previous review have been sufficiently been dealt with (in fact, they were answered very thoroughly)

Reviewer #3: Suggested corrections to revision by Baade:

The revised version of the manuscript has clearly improved, as the introduction is more balanced and many of the experimental results better explained, or at least interpreted so that it fits the model of the authors. 

However there is still a conceptual issue that is not completely resolved. Considering the fundamental nature of the paxillin family adapters and the strong sequence conservation in the LIM domains, it is surprising to note that the very similar cytoplasmic tails of beta1 and beta3 integrins show differences in the mode of paxillin recruitment. Since the identified RGT sequence in the beta3 is not conserved in beta1 integrins, it needs to be stated that the paxillin binding site found in this manuscript, may not reflect a typical paxillin recruitment mechanisms, but rather a specific feature of beta3 integrin. 

Previously a beta3 integrin specific recruitment of the src kinase has been identified by Shattil and coworkers, involving the exact same RGT-motif in beta3 integrin. As this RGT-motif is critical for beta3-mediated recruitment of src for potentially inducing cell adhesion and spreading, it is possible that some of the observed cell spreading defects observed by the authors are due to a failure of beta3 to recruit src and not paxillin. 

When analyzing the beta3-del3 mutation, the reduction in paxillin binding is partial, as is the recruitment of the paxillin LIM3-4A mutation. This translates into partial defects in spreading in beta3 and paxillin ko backgrounds. A more severe phenotype is observed upon deletion of the last 8 amino acids, which fits with the binding site of the kindlin adapter in the beta3 integrin, as well as the more extensive integrin beta3 binding site for paxillin.

In the absence of kindlin, the 3-amino acid deletion in beta3 is sufficient to completely perturb the beta3 induced cell adhesion and spreading. The enhanced severity of the phenotype in kindlin-ko cells shows the critical role of kindlin in the spreading process. 

As the src binding site is much more confined to the RGT-sequence and not spanning 8-10 amino acids of the c-terminal end of the beta3 integrin, it is well possible that the failure of src binding to the deleted RGT sequence contributes to the strong spreading phenotype in the beta3 deletion mutants in the kindlin ko-background. 

Currently information on the src-binding to the RGT-sequence are missing from the manuscript. As the below mentioned reports propose an alternative, paxillin independent explanation for the phenotypes of the beta3-deletion mutants they should be discussed. I would propose that the authors add a short paragraph (for example at line 425 in the discussion), and to mention that the last three amino acids specific to beta3 integrins "RGTcooh" have been shown to bind to the SH3 domain of Src, providing a possible mechanism to provide an integrin beta3-specific ability to induce spreading in the absence of kindlin. (Xiao et al. Blood, 2013, 121, 700-706; Katyal et al. Protein Science, 2013, 22, 1358-1365).

Minor issues : 

Line 126: should read Ripamonti et al., 2021

Line 271: The authors should indicate what changes in the buffer modified these residual peaks in the 4A-mutant (salt, and or pH?). 

Line 277: Please introduce the citation

---

## [Editor Report · Decision Letter 3]

17 Jul 2024

Dear Dr Hauck,

On behalf of my colleagues and the Academic Editor, Carole Parent, I am pleased to say that we can accept your manuscript for publication, provided you address any remaining formatting and reporting issues. These will be detailed in an email you should receive within 2-3 business days from our colleagues in the journal operations team; no action is required from you until then. Please note that we will not be able to formally accept your manuscript and schedule it for publication until you have completed any requested changes.

Please note that I have taken the liberty of adding the information about the Zenodo deposition in the Data Availability Statement in the online submission form. During the production process, please feel free to take a look at the included sentence to make sure you are happy with it before publication. 

PRESS

Best wishes, 

Richard

Richard Hodge, PhD

rhodge@plos.org

PLOS
